

# Bias characterization of OMI HCHO columns based on FTIR and aircraft measurements and impact on top-down emission estimates

Jean-François Müller[1], Trissevgeni Stavrakou[1], Glenn-Michael Oomen[1], Beata Opacka[1], Isabelle De Smedt[1], Alex Guenther[2], Corinne Vigouroux[1], Bavo Langerock[1], Carlos Augusto Bauer Aquino[3], Michel Grutter[4], James Hannigan[5], Frank Hase[6], Rigel Kivi[7], Erik Lutsch[8], Emmanuel Mahieu[9], Maria Makarova[10], Jean-Marc Metzger[11], Isamu Morino[12], Isao Murata[13], Tomoo Nagahama[14], Justus Notholt[15], Ivan Ortega[5], Mathias Palm[15], Amelie Röhling[6], Wolfgang Stremme[4], Kimberly Strong[8], Ralf Sussmann[16], Yao Té[17], and Alan Fried[19-1]

[1]Royal Belgian Institute for Space Aeronomy (BIRA-IASB), Brussels, Belgium
[2]University of California Irvine, Irvine, CA, USA
[3]Instituto Federal de Educaçao, Ciência e Tecnologia de Rondônia (IFRO), Porto Velho, Brazil
[4]Instituto de Ciencias de la Atmósfera y Cambio Climático, Universidad Nacional Autónoma de México (UNAM), Mexico City, México
[5]Atmospheric Chemistry, Observations & Modeling, National Center for Atmospheric Research (NCAR), Boulder, CO, USA
[6]Karlsruhe Institute of Technology (KIT), Institute for Meteorology and Climate Research (IMK-ASF), Karlsruhe, Germany
[7]Finnish Meteorological Institute (FMI), Sodankylä, Finland
[8]Department of Physics, University of Toronto, Toronto, Canada
[9]Institut d'Astrophysique et de Géophysique, Université de Liège, Liège, Belgium
[10]Saint Petersburg State University, Atmospheric Physics Department, St. Petersburg, Russia
[11]Observatoire des Sciences de l'Univers Réunion (OSU-R), UMS 3365, Université de la Réunion, Saint-Denis, France
[12]National Institute for Environmental Studies (NIES), Tsukuba, Ibaraki 305-8506, Japan
[13]Graduate School of Environment Studies, Tohoku University, Sendai 980-8578, Japan
[14]Institute for Space-Earth Environmental Research (ISEE), Nagoya University, Nagoya, Japan
[15]Institute of Environmental Physics, University of Bremen, Bremen, Germany
[16]Karlsruhe Institute of Technology, IMK-IFU, Garmisch-Partenkirchen, Germany
[17]LERMA-IPSL, Sorbonne Université, CNRS, Observatoire de Paris, PSL Université, 75005 Paris, France
[18]Institute of Arctic and Alpine Research, University of Colorado, Boulder, CO, USA

**Correspondence:** Jean-François Müller (jfm@aeronomie.be)

**Abstract.**

Spaceborne formaldehyde (HCHO) measurements constitute an excellent proxy for the sources of non-methane volatile organic compounds (NMVOCs). Past studies suggested substantial overestimations of NMVOC emission in state-of-the-art inventories over major source regions. Here, the QA4ECV (Quality Assurance for Essential Climate Variables) retrieval of

HCHO columns from OMI (Ozone Monitoring Instrument) are evaluated against (1) FTIR (Fourier-transform infrared) column observations at 26 stations worldwide, and (2) aircraft in situ HCHO concentration measurements from campaigns conducted over the U.S. in 2012–2013. Both validation exercises show that OMI underestimates high columns and overestimates low columns. The linear regression of OMI and aircraft-based columns gives $\Omega_{\mathrm{OMI}} = 0.651\,\Omega_{\mathrm{airc}} + 2.95 \cdot 10^{15}$ molec.cm$^{-2}$, with $\Omega_{\mathrm{OMI}}$ and $\Omega_{\mathrm{airc}}$ the OMI and aircraft-derived vertical columns, whereas the regression of OMI and FTIR data gives

$\Omega_{\mathrm{OMI}} = 0.659\,\Omega_{\mathrm{FTIR}} + 2.02 \cdot 10^{15}$ molec.cm$^{-2}$. Inverse modelling of NMVOC emissions with a global model based on OMI





columns corrected for biases based on those relationships leads to much-improved agreement against FTIR data and HCHO concentrations from 11 aircraft campaigns. The optimized global isoprene emissions ($\sim 445\,\mathrm{Tg\,yr^{-1}}$) are 25% higher than those obtained without bias correction. The optimized isoprene emissions bear both striking similarities and differences with recently published emissions based on spaceborne isoprene columns from the CrIS (Cross-track Infrared Sounder) sensor.

Although the interannual variability of OMI HCHO columns is well understood over regions where biogenic emissions are dominant, and the HCHO trends over China and India clearly reflect anthropogenic emission changes, the observed HCHO decline over the Southeastern U.S. remains imperfectly elucidated.

# 1   Introduction

The atmospheric oxidation of non-methane volatile organic compounds exerts multiple influences on tropospheric ozone (e.g.

Chameides et al., 1988; Archibald et al., 2020), fine particulate matter (e.g. Spracklen et al., 2011; Miao et al., 2021) and the oxidizing capacity of the atmosphere (e.g. Seinfeld and Pandis, 1988; Martinez et al., 2003; Bates and Jacob, 2019). Quantifying those impacts is made difficult by the large number and diversity of NMVOCs, with chemical lifetimes ranging from a few minutes to several months (Shu and Atkinson, 1994; Atkinson, 2000) and by important gaps in our understanding of their emissions and chemical degradation mechanisms. Even biogenic isoprene, the most abundantly emitted NMVOC at

the global scale, has very uncertain emissions due (among others) to the strong variability of emission rates among different plant species (Guenther et al., 2006) and to the large uncertainties in the climate and vegetation maps used to calculate the fluxes in state-of-the-art emission models (Arneth et al., 2011; Sindelarova et al., 2014). In addition, field campaigns in various environments such as cities (e.g. Karl et al., 2018) and remote areas (e.g. Read et al., 2012; Wang et al., 2012; Lawson et al., 2015; Travis et al., 2020) suggest that current inventories of anthropogenic and natural emissions of many NMVOCs, and

particularly oxygenated volatile organic compounds (OVOCs) are incomplete. The comparison of measured total OH reactivity with the sum of contributions from measured individual compounds over forests has revealed the presence of a "missing OH reactivity" that is partly, but not always completely, explained by unobserved oxidation products of known VOC precursors (Di Carlo et al., 2004; Sinha et al., 2010; Yang et al., 2016; Nölscher et al., 2016; Sanchez et al., 2021), thereby providing additional indication that primary emissions are missed in emission inventories. For biomass burning as well, the contribution

of unidentified organic compounds to the total NMVOC fluxes might be of the same order, or even larger, than the explicitly identified compounds (Akagi et al., 2011; Andreae, 2019).

Since chemistry transport models do not include unidentified NMVOCs, they are expected to underestimate the total NMVOC flux. Nevertheless, chemistry-transport models (CTMs) using state-of-the-art emission inventories (Guenther et al., 1995, 2006) were found to overestimate the column abundances of formaldehyde, a major NMVOC oxidation product, against

retrieved columns from spaceborne UV-visible sounders such as the Scanning Imaging Absorption Spectrometer for Atmospheric Chartography/Chemistry (SCIAMACHY, Stavrakou et al., 2009; Barkley et al., 2011) and the Ozone Monitoring Instrument (OMI, Millet et al., 2008; Barkley et al., 2011; Fortems-Cheiney et al., 2012; Stavrakou et al., 2015; Bauwens et al., 2016). Furthermore, the largest model biases (exceeding $-40\%$) were found over high-emission areas such as Amazonia,





the Central African rainforest, and the Southeastern U.S. (Barkley et al., 2011; Marais et al., 2014; Bauwens et al., 2016),

where isoprene oxidation is believed to be by far the largest source of formaldehyde (Palmer et al., 2003; Stavrakou et al., 2009). Note that earlier studies using HCHO retrievals from the pioneering Global Ozone Monitoring Experiment (GOME, Palmer et al., 2003; Abbot et al., 2003; Shim et al., 2005; Palmer et al., 2006; Fu et al., 2007; Stavrakou et al., 2009) led to mixed conclusions with both model underestimations and overestimations, primarily due to large differences between different retrievals (De Smedt et al., 2008; Stavrakou et al., 2009). We restrict the following discussion to studies based on subsequent,

higher-quality sounders, in particular OMI.

     The general model overestimation of spaceborne HCHO abundances over source regions was mainly attributed to an over-estimation of isoprene emissions estimated using the Model of Emissions of Gases and Aerosols from Nature (MEGAN, Guenther et al., 2006, 2012), and inverse modelling of emissions – i.e. the derivation of improved emissions constrained by observational data while accounting for the error covariances of the data and the a priori emissions – has led to a reduction

of biogenic emissions globally (e.g., Bauwens et al., 2016). The overestimation of isoprene emissions was also concluded in several studies based on model comparisons with isoprene concentration measurements, in particular over tropical ecosystems (e.g., Houweling et al., 1998; Bey et al., 2001). However, those comparisons are uncertain, due (among others) to the strong variability of isoprene concentrations and to their dependence on OH radical levels, likely too low in those models due to the neglect of OH-recycling mechanisms which were not yet identified (Lelieveld et al., 2008; Paulot et al., 2009; Peeters

et al., 2014; Wennberg et al., 2018; Novelli et al., 2020). A more recent evaluation indicates a model underestimation against aircraft measurements of many reactive VOCs, including formaldehyde and isoprene, over North America (Chen et al., 2019). In addition, recent isoprene flux measurements by eddy covariance from airborne platforms did not suggest overestimations of MEGAN-calculated emissions over the Amazon forest (Gu et al., 2017), the Southeastern U.S. (Yu et al., 2017), and California (Misztal et al., 2014).

Whereas validation studies of the Harvard GOME retrieval showed a good consistency of spaceborne columns against aircraft in situ observations (Martin et al., 2004; Millet et al., 2006), evaluation of OMI HCHO columns against three aircraft campaigns (Boeke et al., 2011) also indicated a good agreement ($-3\%$ bias) with respect to the mean aircraft-based columns, but displayed a larger relative bias ($-17\%$) for higher columns ($> 5 \times 10^{15}\,\mathrm{molec.\,cm^{-2}}$). The latter was tentatively attributed to the preferential sampling of polluted plumes by the aircraft, since the highest HCHO columns were measured in the vicinity

of Mexico City and an airshed near Houston, Texas. However, similar, or even larger, biases were found in the comparison of HCHO columns from several spaceborne sensors, including OMI, against in situ measurements of the SEAC$^4$RS campaign conducted over a wide area covering much of the Southeastern U.S. during summer, a region with high HCHO abundances of primarily biogenic origin (Zhu et al., 2016). The lowest bias ($-20\%$) was realized by the OMI BIRA-V14 retrieval (De Smedt et al., 2015) and could be further reduced (to $-12\%$) when adopting the aircraft vertical shape profiles in the calculation of air

mass factors of the OMI retrieval.

     The first evaluations of HCHO satellite columns using Fourier-transform infrared (FTIR) measurements were conducted in remote environments (Jones et al., 2009; Vigouroux et al., 2009) and suggested a fairly good agreement. In contrast with this, and in qualitative agreement with the results of Boeke et al. (2011), the recent evaluation of TROPOMI (TROPOspheric Moni-





toring Instrument) HCHO columns against FTIR measurements from a network of 25 stations worldwide showed a pronounced
dependence of the bias of TROPOMI columns with the column magnitude, with overestimations (averaging $+25\%$) found for
very low columns ($< 2.5 \times 10^{15}\,\text{molec.\,cm}^{-2}$) and underestimations ($-30.8\%$) for high columns ($> 8 \times 10^{15}\,\text{molec.\,cm}^{-2}$).
The largest underestimation ($-36\%$) was found for the station with the highest average HCHO column, Porto Velho in Ama-
zonia ($29 \times 10^{15}\,\text{molec.\,cm}^{-2}$). Comparable biases can be expected for OMI as for TROPOMI, given the similarity between
the two instruments (De Smedt et al., 2021).

Acknowledging the potentially important consequences of such biases on top-down VOC emission estimates based on
spaceborne HCHO columns, we aim (1) to characterize the biases of HCHO columns from a recent OMI retrieval (De Smedt
et al., 2018) using both FTIR data (using a methodology similar to Vigouroux et al. (2020)) and in situ aircraft measurements
from several campaigns in the U.S., and (2) investigate the consequences of those biases for inverse modelling of NMVOC
emissions based on OMI data.

The manuscript is structured as follows. Section 2 describes the OMI retrieval, the network of FTIR data and the airborne
datasets used in this work; Section 3 presents the validation methodology, the model setup and the emission inversions; Section
4 presents the characterization of OMI biases using FTIR and aircraft data, and proposes a bias-correction formula for use in
inverse modelling; Section 5 presents an assessment of top-down VOC emissions based on OMI, with and without bias correc-
tion; it also provides an evaluation of the optimized results against independent data, and examines the long-term variability
95  and trends of VOC emissions based on the OMI dataset between 2005 and 2016; finally, Section 6 presents the conclusions of
this study.

## 2   Description of observational datasets

### 2.1   The OMI HCHO columns

The Ozone Monitoring Instrument (OMI) was launched in 2004 aboard the Aura mission in a low-Earth polar orbit crossing the
100  Equator around 13:30 LT. OMI is a nadir spectrometer that measures the solar radiation backscattered by the Earth's atmosphere
and surface between 270 and 500 nm (Levelt et al., 2006). OMI has a 2600 km wide swath (divided into 60 across-track rows),
providing near-global coverage every day. Due to an anomaly affecting the CCD detector, an increasing number of rows had to
be filtered out, resulting in gradual degradation of the coverage to about $50\%$ (Torres et al., 2018). The OMI ground pixel size
varies from $13 \times 24\,\text{km}^2$ at nadir to $28 \times 150\,\text{km}^2$ at the edges of the swath.

105  The OMI HCHO dataset (v1.2, https://doi.org/10.18758/71021031) was developed within the EU-FP7 QA4ECV project.
The retrieval algorithm for HCHO is described in De Smedt et al. (2018). It is based on a three-step DOAS (differential optical
absorption spectroscopy) method. First, the fit of the slant columns is performed in a spectral window between 328.5 and
359 nm, with HCHO cross sections from Meller and Moortgat (2000). The slit function of each OMI row is adjusted on
a daily basis as part of the wavelength calibration procedure, and the absorptions cross sections are convolved accordingly.
The DOAS reference spectrum is updated every day with averaged Earth radiances measured in the equatorial Pacific. The
fit therefore provides a differential slant column, which corresponds to the HCHO excess over source regions in comparison



to the remote background. In a second step, the slant columns are converted to tropospheric vertical columns using a lookup table of vertically-resolved air mass factors calculated at 340 nm using the VLIDORT v2.6 radiative transfer model (Spurr, 2008). Surface albedo is obtained from the monthly OMI climatology at 0.5° resolution (Kleipool et al., 2008). Daily a priori

vertical profiles are obtained from the TM5 analysis, at 1° spatial resolution (Williams et al., 2017). For this work, the cloud correction is switched off, except for a strict filtering (CF<0.2, CF being the cloud fraction) based on the Fresco v7 cloud product (Veefkind et al., 2016), and clear-sky air mass factors (AMF) are used in lieu of the cloud-corrected AMFs of the standard QA4ECV product. This choice ensures an optimal consistency with the TROPOMI HCHO dataset (De Smedt et al., 2021). Indeed, the TROPOMI HCHO retrieval is inherited from the QA4ECV algorithm with the aim to generate a consistent

time series of early afternoon observations. Finally, to correct for any global offset and for stripes arising between the rows, a background correction is performed on daily basis using the HCHO slant columns over the Pacific Ocean. To compensate for the background HCHO over the equatorial Pacific, HCHO is obtained from the TM5 model in the reference region. Several diagnostic variables are provided together with the measurements. Besides the cloud filter, the recommended processing is applied, in particular with respect to the row anomaly (De Smedt et al., 2018). For every pixel, column averaging kernels and

a priori profiles are provided, as well as the random and systematic components of the tropospheric column uncertainty.

## 2.2 Ground-based FTIR HCHO data

The FTIR stations participating in the present study are listed in Table S1 (see also Fig. 1(b)). Most stations are affiliated to the Network for the Detection of Atmospheric Composition Change (NDACC). The InfraRed Working Group (IRWG) of NDACC (https://www2.acom.ucar.edu/irwg) requires standardized high-quality instruments, namely high-resolution spec-

trometers mostly from the same manufacturer (Bruker 120HR or 125HR), and homogenized retrieval algorithms: PROFITT9 (Hase et al., 2006) and SFIT4 (Pougatchev et al., 1995). These codes retrieve information on atmospheric composition from solar absorption measurements performed under clear-sky conditions in the infrared spectral region, and provide total columns as well as low-vertical resolution profiles of many atmospheric species.

Harmonized retrieval settings have been set up and used within the whole IRWG in order to build a consistent ground-

based FTIR HCHO data set for robust interpretation of satellite and model validation (Vigouroux et al., 2018). This data set has proven its value for the validation of TROPOMI HCHO (Vigouroux et al., 2020, see also the quarterly reports https://mpc-vdaf.tropomi.eu), GOME-2 (https://acsaf.org/valreps.php) or more recently of OMPS (Kwon et al., 2023). We refer to Vigouroux et al. (2018) for details about the HCHO retrieval settings. The most critical aspects with regard to harmonization within the network are the spectral signatures and the spectroscopic parameters. The HCHO fitted spectral sig-

natures belong to the $\nu_1$ and $\nu_5$ vibrational bands, around 3.6 μm. The spectroscopic database is the atm16 line list from G. Toon (http://mark4sun.jpl.nasa.gov/toon/linelist/linelist.html), which corresponds to HITRAN 2012 (Rothman et al., 2013) for formaldehyde. Low-vertical resolution formaldehyde profiles are obtained using Tikhonov regularization (Tikhonov, 1963). However, the degrees of freedom for signal (DOFS) are low for HCHO (from 1 to 1.6) due to its weak spectral signature, implying that mainly a total column can be retrieved. The sensitivity of the measurement is highest in the free troposphere, but a

good sensitivity is achieved near the surface (see Fig. 4 of Vigouroux et al. (2018)). The FTIR HCHO uncertainty is calculated



according to Rodgers (2000). The systematic uncertainty, mostly due to uncertainty on the spectroscopic parameters, is $\sim 13\%$ (Vigouroux et al., 2018). The random uncertainties can be as low as $1.0 \times 10^{14}$ molec. $\mathrm{cm}^{-2}$ (8%) for clean sites such as Eureka and up to $5.3 \times 10^{14}$ molec. $\mathrm{cm}^{-2}$ (7%) for polluted stations such as Paris. Only the Mexico City site has a larger random uncertainty ($11.1 \times 10^{14}$ molec. $\mathrm{cm}^{-2}$) because it operates a low-resolution spectrometer (Vertex 80). Additionally, note that significant random and systematic smoothing uncertainties also exist (Vigouroux et al., 2018); however, those uncertainties vanish when the FTIR averaging kernels are used to smooth the model profiles, as done in this study (Rodgers and Connor, 2003).

## 2.3 Airborne HCHO data

Table 1 lists the aircraft campaign data used in this study. Among those, four HCHO datasets from three campaigns conducted in 2012-2013 over the United States are used to evaluate OMI HCHO columns, as described in Sect. 3.3. All campaigns are used to evaluate the inverse modelling results. More specifically,

- the DC3 (Deep Convective Clouds and Chemistry) NASA mission sampled the atmospheric composition over the Central U.S. in May-June 2012 (Barth et al., 2015). Formaldehyde mixing ratios were acquired from two aircraft platforms, the NASA DC-8 and the NSF/NCAR Gulfstream V (GV), equipped with similar infrared absorption spectrometers employing difference frequency generation (DFG) laser sources: the DFGAS (Difference Frequency Generation Absorption Spectrometer) instrument on the DC-8 (Weibring et al., 2007) and the more sensitive CAMS (Compact Atmospheric Multispecies Spectrometer) instrument on the GV (Richter et al., 2015). More details regarding the instruments can be found in Fried et al. (2016). A comparison between the DFGAS and CAMS measurements indicates a very good agreement between the two instruments, with the DGFAS measurements being slightly higher (by 11%, Fried et al. (2016)).

- the SEAC$^4$RS (Studies of Emissions, Atmospheric Composition, Clouds and Climate Coupling by Regional Surveys) campaign was conducted over the southeastern U.S. in August-September 2013 on board the NASA DC-8 aircraft (Toon et al., 2016). The mission took place place over regions rich in biogenic VOC emissions. Two different instruments were used: CAMS and the ISAF (In situ Airborne Formaldehyde) instrument (Cazorla et al., 2015). The ISAF data were found to be about 10% higher than the CAMS values (Zhu et al., 2016). Only the CAMS data are used here.

- the DISCOVER-AQ (Deriving Information on Surface conditions from Column and Vertically Resolved Observations Relevant to Air Quality) was a multi-year aircraft mission led by NASA (Crawford and Pickering, 2014). The HCHO measurements were acquired using the DFGAS technique (Weibring et al., 2007). The campaign over California (January-February 2013) is used in the OMI HCHO validation. The campaigns over Maryland (2011) and Colorado (2014) are used for model evaluation.

- the INTEX (Intercontinental Chemical Tranport Experiment) campaigns took place in 2004 and 2006. We use here the measurements of the March 2006 INTEX-B campaign, also named MILAGRO (Megacity Initiative: Local and Global





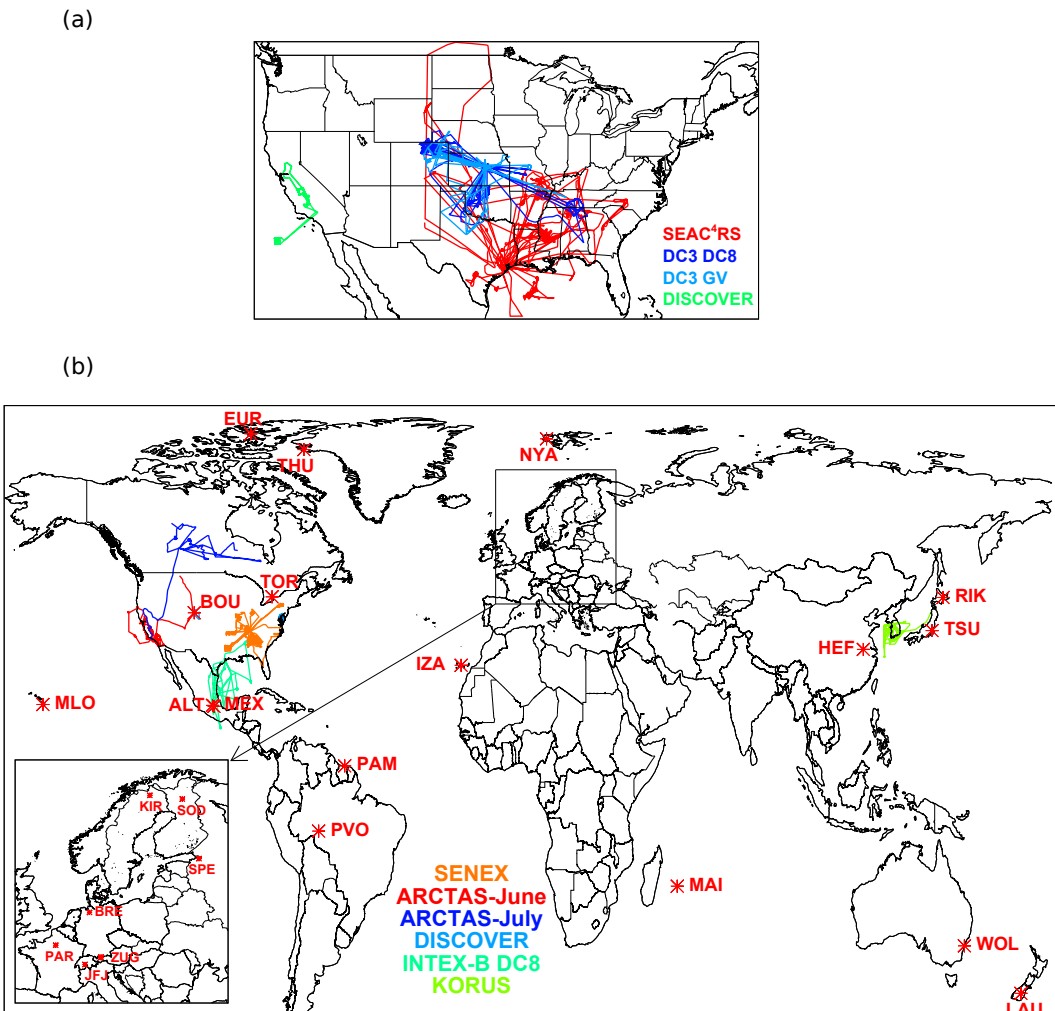

**Figure 1.** (a) Flight tracks of the aircraft missions (DC3, SEAC[4]RS, DISCOVER-AQ California) used as constraints in the aircraft-based inversion, (b) flight tracks of the additional aircraft campaigns used for global model evaluation, and location of FTIR stations used in this work to evaluate OMI HCHO columns. The station coordinates and codes are provided in Table S1. Note that the stations Garmisch and St.Denis are not shown, given their close proximity to the stations Zugspitze (ZUG) and Maïdo (MAI), respectively.

Research Observations) (Molina et al., 2010). The NASA DC8 measured chemical composition over Mexico, Texas and the Gulf of Mexico below ca. 10 km altitude. HCHO was measured using Tunable Diode Laser Spectroscopy (TDLAS) (Fried et al., 2008). The INTEX-B campaign of April-May 2006 is not used here, as it was conducted primarily over the Pacific Ocean, where HCHO levels are insensitive to emissions over land.

180




**Table 1.** Aircraft campaign datasets used in this work for OMI and model evaluation. Datasets 1-4 are used to characterize OMI biases. All datasets are used to evaluate the top-down emission inversions.

| Dataset number | Aircraft dataset | Period | Measurement technique | Reference |
|---|---|---|---|---|
| 1 | DC3 (DC8) | May-June 2012 | DFGAS | Weibring et al. (2007) |
| 2 | DC3 (GV) | May-June 2012 | CAMS | Richter et al. (2015) |
| 3 | SEAC⁴RS | Aug.-Sep. 2013 | CAMS | Richter et al. (2015) |
| 4 | DISCOVER-AQ California | Jan.-Feb. 2013 | DFGAS | Weibring et al. (2007) |
| 5 | MILAGRO (DC8) | March 2006 | TDLAS | Fried et al. (2008, 2011) |
| 6 | ARCTAS-CARB | June 2008 | TDLAS | Fried et al. (2008) |
| 7 | ARCTAS-B | July 2008 | TDLAS | Fried et al. (2008) |
| 8 | DISCOVER-AQ Maryland | July 2011 | DFGAS | Weibring et al. (2007) |
| 9 | DISCOVER-AQ Colorado | July-Aug. 2014 | DFGAS | Weibring et al. (2007) |
| 10 | SENEX | June-Jul 2013 | ISAF | Cazorla et al. (2015) |
| 11 | KORUS-AQ | May-June 2016 | CAMS | Richter et al. (2015); Fried et al. (2020) |

- the ARCTAS (Arctic Research of the Composition of the Troposphere from Aircraft and Satellites) campaigns took place in 2008 (Jacob et al., 2010). Whereas the ARCTAS-A campaign (not used here) targeted the springtime atmospheric composition over the Arctic, the ARCTAS-CARB (June) and ARCTAS-B (July) campaigns sampled the tropospheric composition during the summer between 34 and >80° N. Only data below 60° N are used here. The TDLAS technique (see above) was employed to measure HCHO during both missions.

- the Southeast Nexus (SENEX) campaign (Warneke et al., 2016) used the NOAA WP-3D aircraft to sample the lower troposphere (below ca. 6 km altitude) over the Southeast U.S. in June 2013. HCHO was measured using the ISAF instrument (see above).

- the KORUS-AQ (Korea-United States Air quality) campaign investigated air composition over Korea and surrounding areas in May-June 2016. Formaldehyde was measured throughout the troposphere by the NASA DC8 using CAMS (see above).

For all campaigns, we exclude HCHO measurements from urban plumes (identified as $[NO_2] > 4$ ppbv or $[NO_x]/[NO_y] > 0.4$) and biomass burning plumes ($[CH_3CN] > 225$ pptv), as well as nighttime data (before 9 AM or after 18 PM local time) and measurements over oceans. Data above 4 km altitude are also excluded. Furthermore, as the average HCHO mixing ratio from the CAMS instrument were about 13% lower than those of the ISAF instrument during SEAC⁴RS, we increase the CAMS values by 6.5% in order to bring the SEAC⁴RS data midway between CAMS and ISAF, while also reducing the bias between the GV data (CAMS) and DC8 data (DFGAS) from the DC3 mission.



The measurements are publicly available via data archive centers (see Data availability). The flight tracks are shown in Fig. 1.

## 3 Methodology

### 3.1 Validation using FTIR measurements

The validation of OMI HCHO data using the FTIR ground-based network follows the methodology applied for TROPOMI by Vigouroux et al. (2020). Only the collocation criteria are chosen differently, due to the lower precision of OMI compared to TROPOMI columns: we average the OMI pixels within 50 km around each station, which leads to a mean value of 13 pixels per collocation. Only pixels satisfying the recommended quality criteria (Sect. 2.1) are averaged. In addition, the collocated pair is kept only if at least 10 valid OMI pixels are available for averaging.

We apply the formalism of Rodgers and Connor (2003) in order to account for the different a priori vertical profiles in the FTIR and OMI retrievals (single climatological profile from the WACCM model and daily profiles from TM5, respectively) and for the different vertical sensitivity of both instruments. For each FTIR profile in time coincidence (within 3 hours of the OMI overpass time) and for each individual OMI pixel (within 50 km of the station), the OMI a priori profile is substituted for the FTIR profile (see Eq. 2 in Vigouroux et al., 2020), and the corrected FTIR profile is smoothed after regridding (and extrapolation if needed) to the satellite grid using the OMI averaging kernel (Eq. 3 of Vigouroux et al., 2020). The vertical re-gridding ensures that the same upper boundary for the tropospheric column definition is used in both products (the one from OMI). In addition, a scaling factor is applied to the OMI and smoothed FTIR columns to correct for the altitude difference between the OMI pixel and the the station (Eq. 4 of Vigouroux et al., 2020). That factor is taken as the fraction of the a priori OMI column that lies above the station altitude.

In summary, each coincident pair consists of the average of the smoothed and scaled FTIR columns within 3 hours of the OMI overpass and the average of the individually-scaled OMI pixels within 50 km of the station.

### 3.2 HCHO simulation using the MAGRITTEv1.1 CTM

The Model of Atmospheric composition at Global and Regional scales using Inversion Techniques for Trace gas Emissions (MAGRITTE v1.1) is a chemistry-transport model based on the previous IMAGES (Intermediate Model for the Annual and Global Evolution of Species) model (Muller and Brasseur, 1995; Stavrakou et al., 2018). MAGRITTE v1.1 calculates the distribution of 182 chemical species, among which 141 compounds undergo transport (advection, deep convection, and turbulent mixing in the boundary layer) in the model. The chemical mechanism includes a detailed description of isoprene and other biogenic volatile organic compounds oxidation (BVOCs) mechanisms (Müller et al., 2019). In particular, it incorporates an up-to-date representation of isoprene peroxy radical unimolecular reactions and other recent mechanistic advances relevant to BVOC oxidation (Müller et al., 2019). The photolysis rates are interpolated from tabulated values calculated using the TUV photolysis estimation package (Madronich and Flocke, 1998). Most model parameterizations, including the chemical mechanism for anthropogenic and pyrogenic organic compounds are obtained from the IMAGES model (Stavrakou et al.,



2009; Bauwens et al., 2016). MAGRITTE can be run either as a global model, at $2° \times 2.5°$ resolution, or as regional model, at $0.5° \times 0.5°$ resolution. In regional mode, the boundary conditions of the lateral borders are provided off-line by the global model. The chemical concentrations are calculated on a $\sigma$-pressure coordinate grid, with 40 vertical levels distributed within the troposphere and the lower stratosphere (below the 44 hPa level).

Meteorological fields are obtained from the ERA5 ECMWF reanalyses (Hesbach et al., 2020). The effect of diurnal variation on the photolysis rates and kinetic rate constants are taken into account through correction factors calculated from model simulations with a 20-min time step. These correction factors are used to calculate the diurnal cycle of HCHO columns required for the comparisons with ground-based and aircraft measurements.

Anthropogenic emissions of CO, NOx, $SO_2$ and carbonaceous aerosols are taken from the HTAPv2 (Hemispheric Transport of Air Pollution version 2) inventory (Janssens-Maenhout et al., 2015) for 2010. The anthropogenic NOx emissions over the U.S. are adjusted (Travis et al., 2016; Müller et al., 2019) to match observed $NO_x$ concentration and $HNO_3$ deposition data. The speciated emissions of NMVOCs are obtained from the EDGARv4.3.2 inventory (Huang et al., 2017) between 2005 and 2012, and are taken equal to their 2012 values afterwards. The global annual anthropogenic NMVOC source is estimated at 162.3 Tg in 2005 and increases annually by $> 1\%$ to reach 179.8 Tg in 2012. Vegetation fire emissions are provided from the GFED4s database (van der Werf et al., 2017), with vertical injection profiles from Sofiev et al. (2013). The global biomass burning flux is estimated to range between 78 and 107 $Tg\,yr^{-1}$, and the average global flux over 2005–2017 was 90.5 $Tg\,yr^{-1}$. Isoprene and monoterpene fluxes are calculated by the MEGAN model embedded in the MOHYCAN canopy environment model (Müller et al., 2008; Guenther et al., 2012; Bauwens et al., 2018) at $0.5° \times 0.5°$ based on the ERA5 reanalysis meteorological fields and Leaf Area Index (LAI) data from MODIS Collection 6 reprocessed by Yuan et al. (2011). The $CO_2$ inhibition effect is accounted for using the parameterization of Possell and Hewitt (2011). The effects of soil moisture stress are neglected, since previous model evaluations against OMI data have shown a deterioration of temporal correlation when accounting for the soil moisture activity factor calculated using MEGANv2.1 (Guenther et al., 2006) and ECMWF soil moisture fields (Bauwens et al., 2018; Stavrakou et al., 2018). The global annual isoprene flux ranges between 414 Tg (in 2008) and 452 Tg (in 2016). The average annual flux amounts to 433 $Tg\,yr^{-1}$ over 2005-2017. Biogenic methanol is also calculated according to MEGAN as described in Stavrakou et al. (2011). The monthly fluxes of isoprene and methanol emissions are available online at http://emissions.aeronomie.be. The annual monoterpene fluxes range between 109 Tg (in 2008) and 120 Tg (in 2016). Biogenic acetaldehyde and ethanol emissions (amounting to 22 $Tg\,yr^{-1}$ for each compound) are calculated as described in Millet et al. (2010). Biogenic emissions of $C_2H_4$ (scaled to 4 $Tg\,yr^{-1}$ globally), HCHO (4 $Tg\,yr^{-1}$) and $CH_3COCH_3$ (28 $Tg\,yr^{-1}$) are also obtained from MEGAN (Guenther et al., 2012) (http://eccad.aeris-data.fr). Finally, oceanic emissions are included for acetaldehyde (56 $Tg\,yr^{-1}$), methanol (49 $Tg\,yr^{-1}$) and acetone (63 $Tg\,yr^{-1}$) (Müller et al., 2018).

## 3.3 Aircraft-based inversion

The regional MAGRITTE model with its inverse modelling capability is used to generate HCHO model distributions that closely approximate the aircraft observations from the campaigns DC3, SEAC[4]RS and DISCOVER-AQ California (Table 1).



This is realized by adjusting the NMVOC emissions used in the model and minimizing a cost function which quantifies the
overall discrepancy between the model-calculated mixing ratios and the observations. The cost ($J$) is expressed as

$$J(\mathbf{f}) = \frac{1}{2}\left[(H(\mathbf{f}) - \mathbf{y})^T \mathbf{E}^{-1}(H(\mathbf{f}) - \mathbf{y}) + \mathbf{f}^T \mathbf{B}^{-1}\mathbf{f}\right],\tag{1}$$

where $\mathbf{f}$ denotes the vector of dimensionless emission parameters, $H(\mathbf{f})$ is the chemistry-transport model operating on the
control variables, $\mathbf{y}$ is the observation vector, and $\mathbf{E}$, and $\mathbf{B}$ are the covariance matrices of the errors on the observations and
the emission parameters $\mathbf{f}$, respectively. For each observation, the model is sampled at the same day, hour, pixel and altitude as
the measurement. However, the observation vector $\mathbf{y}$ and its model counterpart $H(\mathbf{f})$ consist of campaign-averaged (observed
or modelled) mixing ratios in each model pixel ($0.5°\times0.5°$) for which observations are available. In this way, different model
pixels contribute similarly to the cost function, despite the spatially heterogeneous sampling of air composition by the aircraft.
In this way, we avoid giving excessive weight to intensively surveyed areas (e.g. the Houston ship channel during the SEAC[4]RS
campaign, Toon et al., 2016), since those areas were often chosen due to special features (e.g. high pollution levels) and might
not be representative at a larger scale.

The monthly averaged emission flux for a given category (anthropogenic, pyrogenic or biogenic) is expressed as

$$G(\mathbf{x}, t, \mathbf{f}) = \sum_{j=1}^{m}\exp(f_j)\phi_j(\mathbf{x}, t)\tag{2}$$

with $\phi_j$ the a priori emission distributions detailed in Section 3.2. The emission for a given category and pixel is not optimized
when its maximum monthly value over the course of the year is lower than $10^9$ molec. $\mathrm{cm}^{-2}\,\mathrm{s}^{-1}$. This threshold is sufficiently
low that the emission of most pixels are optimized over the contiguous U.S. for the biogenic and anthropogenic categories. The
total number of optimized parameters is $\sim 2.2 \times 10^5$.

The matrix $\mathbf{E}$ is assumed diagonal, and includes instrumental errors as well as representativity and model errors. The total
uncertainty is derived by quadrature addition of a $15\%$ relative uncertainty and a $200$ pptv absolute error. The $15\%$ error is
slightly higher than both the estimated instrumental systematic uncertainty (e.g., $12.4\%$ for the TDLAS instrument) (Fried et
al., 2008) and the typical bias between different measurement techniques (see above). The $200$ pptv contribution is higher than
the limit of detection, typically lower than $100$ pptv (Fried et al., 2008) but is intended to give more weight to higher HCHO
mixing ratios in the cost function.

The diagonal elements of $\mathbf{E}$ are taken equal to $1.1^2$, i.e. the errors on all emission parameters are assumed to be a factor 3
($e^{1.1}$). Anthropogenic emission parameters from pixels in the same country are weakly correlated (coefficient of 0.1), whereas
parameters for different countries are assumed uncorrelated. For biogenic and pyrogenic emissions, a decorrelation length of
$100$ km is used. The cost function is minimised using an quasi-Newton optimization algorithm involving the calculation of the
gradient of the cost function by the adjoint of the model (Müller and Stavrakou, 2005). The convergence criterion is a reduction
of the norm of the gradient of the cost $J$ by an order of magnitude. Typically, this criterion is reached after 20 iterations.

The model domain includes the contiguous U.S. (10-54°N, 65-130°W). Simulations start on July $1^{st}$ 2011 and last 2.5 years.

The optimized HCHO distributions are used to calculate, for each campaign, a campaign-average gridded HCHO column
field accounting for the sampling times and averaging kernels of the OMI retrievals. Those columns are then compared to the




corresponding OMI columns at the locations of the aircraft measurements aggregated onto the model grid. Model pixels with less than 10 OMI measurements, or less than 5 aircraft measurements below 4 km are excluded from analysis.

## 3.4 Inversion constrained by satellite columns

The methodology for optimizing global NMVOC emissions based on OMI data is similar to that presented in the previous
section, except (1) the global model is used, (2) monthly-averaged bias-corrected OMI HCHO columns binned onto the model resolution ($2° \times 2.5°$) are used as contraints, (3) a decorrelation length of 300 km is assumed for a priori error correlations in the biogenic and pyrogenic sectors, to account for the coarser model resolution (4) separate inversions are performed for each year between 2005 and 2017, and each simulation starts on July $1^{st}$ of the year preceding the target simulation year. For each optimization, the number of optimized parameters is $\sim 10^5$. The convergence criterion (reduction by a factor of 30 of the norm
of the gradient of $J$) is attained after typically $15 - 20$ iterations.

The OMI column uncertainty is obtained by quadrature addition of the OMI retrieval uncertainty and an absolute error taken to be $2 \times 10^{15}$ molec. cm$^{-2}$. The retrieval error consists of a systematic and a random component, but the latter is greatly reduced upon averaging. In order to limit the noise, monthly averages based on less than 20 valid measurements are excluded from analysis. The systematic uncertainty is typically $35 - 55\%$ over source areas in tropical regions and during summertime
at mid-latitudes. In winter at mid-latitudes, the retrieval error usually exceeds $80\%$. The OMI monthly averages are compared to the corresponding MAGRITTE monthly averages. Those are calculated from daily values accounting for the number of measurements and averaging kernel for each day (also binned onto the model resolution) and for the sampling time ($\sim$13h30 LT) of observation at each location.

## 4 Results

### 4.1 OMI HCHO bias characterization using FTIR data

Due to the higher noise of OMI data (compared to TROPOMI), a relatively poor correlation is obtained between individual coincident pairs, with a Pearson's coefficient of 0.55, whereas a correlation coefficient of 0.81 was found in the evaluation of TROPOMI data against FTIR data (Vigouroux et al., 2020). We use therefore the monthly means of coincident pairs to derive a more robust linear relationship between OMI and FTIR columns. The scatter plot of the coincident monthly-averaged FTIR
and OMI columns is shown in Fig. 2. The comparison of monthly means shows a Pearson's correlation coefficient of 0.67. The regression using the Theil-Sen estimator (Sen, 1968) yields a positive OMI constant bias (intercept of $2.02 \times 10^{15}$ molec. cm$^{-2}$) and a negative OMI proportional bias (slope of 0.659). The positive OMI bias for clean sites and negative bias for polluted sites is similar as for TROPOMI validation (Vigouroux et al., 2020).





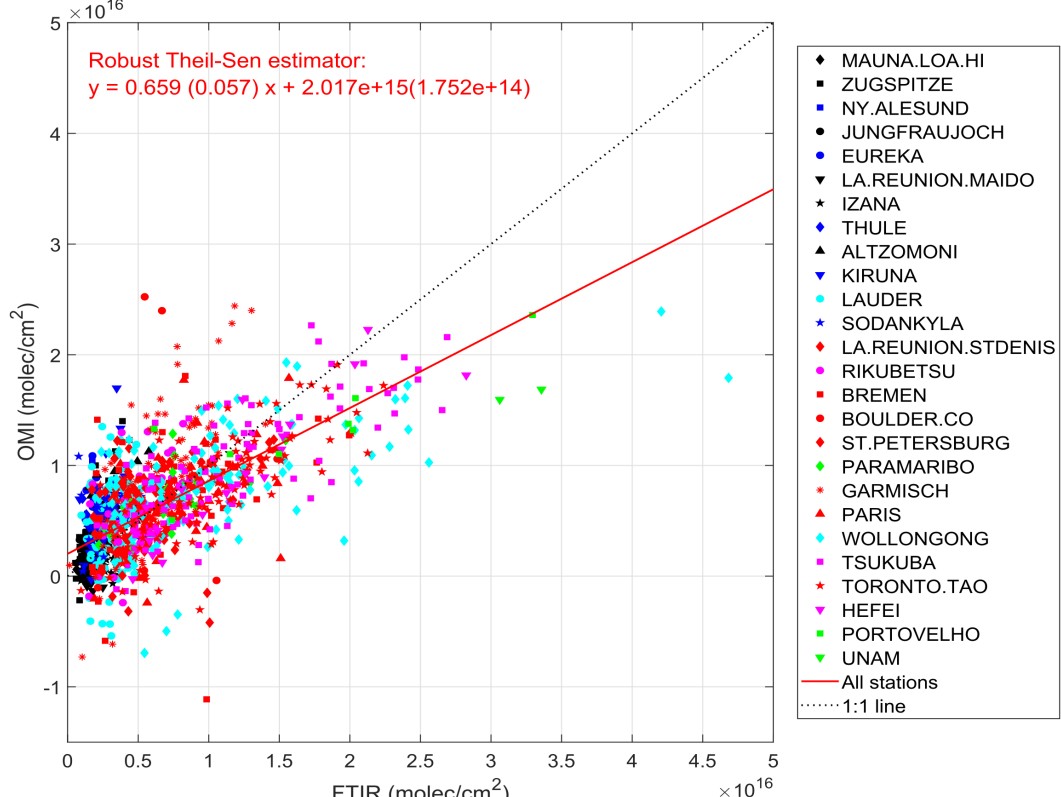

**Figure 2.** Scatter plot of co-located OMI and FTIR HCHO monthly columns over 2005–2017. The red line represents the Theil-Sen regression. Its slope and intercept (with their 1-$\sigma$ errors between brackets) are given inset in red.

## 4.2 OMI HCHO bias characterization using aircraft data

Here the OMI data are being evaluated against aircraft in situ measurements, using the regional MAGRITTE model as transfer standard. Figure 3(a-c) displays the distribution of observed HCHO mixing ratios from the three aircraft campaigns. The highest concentrations ($> 3\,\mathrm{ppbv}$) are seen in the Southeast during summertime, whereas lower, but still significant levels ($1 - 3\,\mathrm{ppbv}$) are observed over the Central U.S. (Colorado, Nebraska) during summer and over California in winter. The model with a priori emissions reproduces very well the high summer values over the Southeast, with very little bias found against SEAC[4]RS data

(Fig. 3(d-f)). Over California and Central U.S., substantial model underestimations are found, reaching typically a factor of 2. Similar results were obtained by Chen et al. (2019) with the GEOS-Chem model, indicating a good model agreement in regions dominated by biogenic VOC emissions according to MEGAN (i.e. the Southeast U.S.), and large underestimations over California and the Central U.S., as shown by extensive comparisons with aircraft campaigns. Not only HCHO, but many VOCs are similarly underestimated in those areas (Chen et al., 2019), suggesting the presence of missing VOC sources in




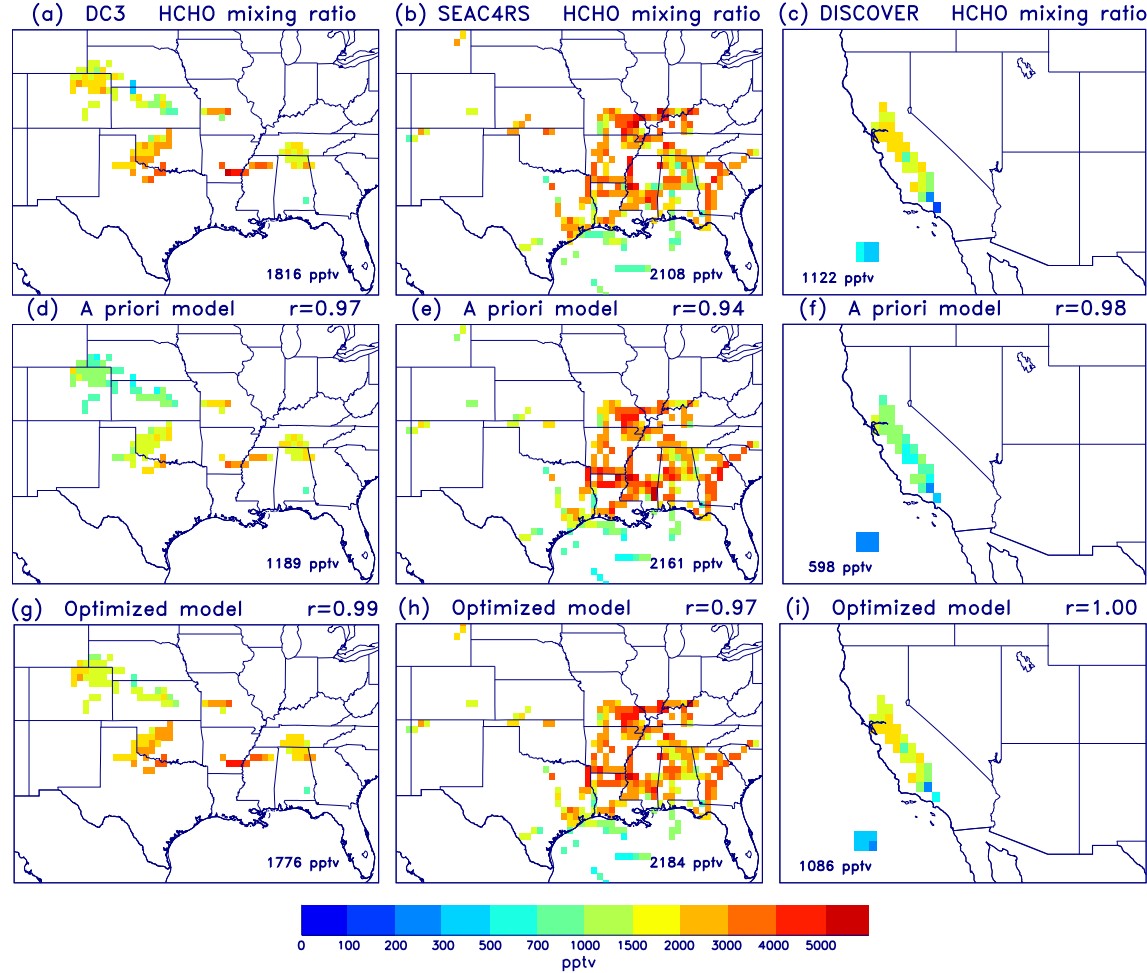

**Figure 3.** Measured HCHO mixing ratio (pptv) below 4 km altitude from the campaigns (a) DC3, (b) SEAC⁴RS, and (c) DISCOVER-AQ California, and corresponding model distributions using a priori emissions (d, e, f) and with optimized emissions (g, h, i). Urban and biomass burning plumes were filtered out as described in the text. The data are gridded to the model resolution. The averaged measured or modelled mixing ratio is given inset for each campaign.

emission inventories. Urban VOC emissions are likely strongly underestimated (Karl et al., 2018), likely partly due to the use of volatile chemical products (VCPs) (McDonald et al., 2018) and the underestimation of urban biogenic VOC emissions (Gu et al., 2021). Over Colorado, Nebraska and Texas (among others), fossil fuel exploitation is a large source of alkanes and other VOCs (Pétron et al., 2014; Franco et al., 2016; Tzompa-Sosa et al., 2019) likely responsible for a large part of the model discrepancy.

In contrast with the prior simulation, the optimized model using adjusted VOC emissions reproduces very well the observed distributions (Fig. 3(g-i)), with spatial correlation coefficients of 0.97 or more for each campaign, and average biases of at





most 3%. This agreement is achieved through a substantial increase of anthropogenic emissions throughout the U.S., whereas biogenic emissions are decreased over much of Southeast U.S. and biomass burning emissions undergo a moderate increase (see Table S2 and Fig. S1 in the Supplement). Since the emission parameters are severely underconstrained by the inversion due

to the poor coverage of the observations over the model domain, the solution found by the inversion has limited reliability and is strongly dependent on the a priori emission distribution and inversion setup, in particular the covariance matrix of the emission parameters. Despite this caveat, the optimization reproduces very well both the HCHO horizontal distribution (Fig. 3(g-i)) and the vertical profile shape of HCHO mixing ratios for all campaign datasets, as seen on Fig. 4. While panels (a-d) in Fig. 4 display the campaign-averaged profiles for the datasets used in the emission inversion, Fig. 4(e) shows the profiles for the

SENEX campaign which took place in 2013 but was not used as constraint in the inversion. Although the observed profile shape is correctly simulated by the model, the overall agreement is significantly lower for this campaign ($-22\%$ bias) than for the campaign datasets used in the inversion. Given the significant overlap of the SENEX and SEAC$^4$RS spatial coverages (see Fig. 1), the underestimation of SENEX HCHO by the model suggests a shift of the summertime biogenic VOC emission peak towards spring, since SENEX and SEAC$^4$RS were conducted (primarily) in June and August, respectively. This finding, which

is in agreement with the previous determination of isoprene emissions based on satellite (GOME) data by Palmer et al. (2006), will be further examined using OMI and the global model in Sect. 5.

For the campaigns used in the inversion, a regression of the observed and simulated concentrations yields a slope of almost 1 and a correlation coefficient of 0.93 (Fig. 5(a)). In contrast with this, Fig. 5(b) shows that the OMI HCHO columns are significantly biased with respect to co-located columns calculated using the optimized model distributions (Sect. 3.3). High OMI

HCHO columns ($>\sim 12 \times 10^{15}$ molec. cm$^{-2}$) are underestimated by up to ca. 20% for columns $\sim 20 \times 10^{15}$ molec. cm$^{-2}$, whereas low columns ($<\sim 8 \times 10^{15}$ molec. cm$^{-2}$) are generally overestimated. This underestimation of high columns is relatively similar with the slight underestimation (12%) of OMI HCHO columns from the BIRA V14 product (De Smedt et al., 2015) against SEAC$^4$RS HCHO data determined by Zhu et al. (2016). It should be stressed, however, that Zhu et al. (2016) filtered out very negative HCHO columns ($< -5 \times 10^{15}$ molec. cm$^{-2}$) from the V14 product, which contributed to increase

the campaign-averaged values and reduce the OMI underestimation. Without this filter, the BIRA V14 data are significantly lower than the QA4ECV columns (see Fig. S2). The underestimation of high columns is also qualitatively consistent with the mean OMI bias of $-17\%$ derived by Boeke et al. (2011) for OMI columns $> 5 \times 10^{15}$ molec.cm$^{-2}$, based on comparisons with several aircraft campaigns conducted in 2008.

A linear regression (Theil-Sen) of OMI and aircraft constrained model columns (Fig. 5(b)) yields

$$\Omega_{\text{OMI}} = 0.651\,\Omega_{\text{airc}} + 2.95 \cdot 10^{15}, \tag{3}$$

where $\Omega_{\text{OMI}}$ and $\Omega_{\text{airc}}$ are the HCHO columns (molec. cm$^{-2}$) from OMI and from the aircraft-constrained model simulation, respectively. This regression is shown as a black line in Fig. 5(b) and compared with the result of the regression based on FTIR data (see Sect. 4.1), shown as a dashed red line. The slopes of the two regressions are almost identical, but the FTIR-based fit has a smaller intercept and suggests a larger negative bias of high OMI columns than the aircraft-based evaluation. For

example, for an OMI column of $18 \times 10^{15}$ molec. cm$^{-2}$, the FTIR and aircraft datasets suggest underestimations by factors




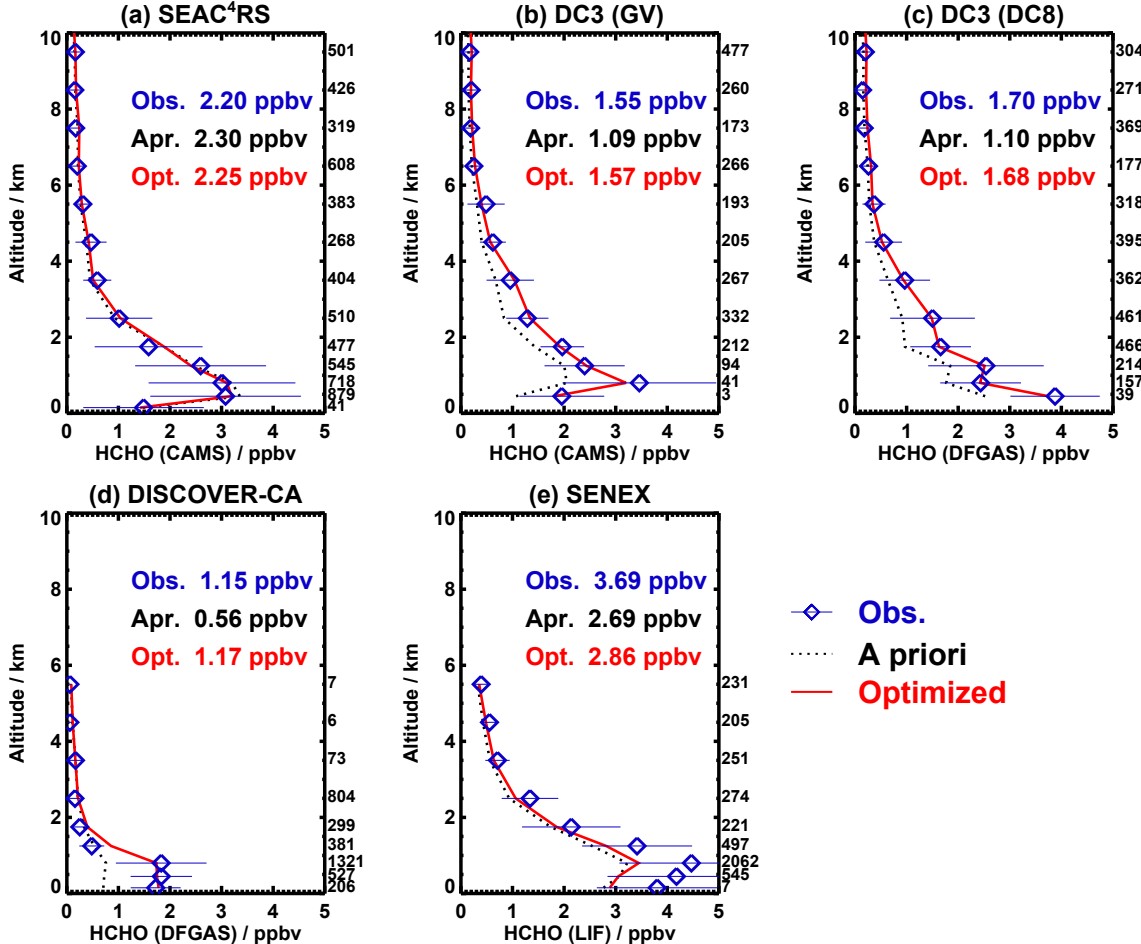

**Figure 4.** Measured and modelled profiles of HCHO mixing ratios over land (in ppbv) for (a) SEAC⁴RS, (b) DC3 (GV), (c) DC3 (DC8), (d) DISCOVER-AQ California, and (e) SENEX. Urban and biomass burning plumes were filtered out as described in the text. The averaged measured and modelled (a priori and optimized) mixing ratios below 4 km altitude are given inset for each dataset. The number of observations are given to the right of each plot. The boundaries of the altitude bins are 0, 0.3, 0.6, 1, 1.5, 2, 3, 4, 5, 6, 7, 8, 9, 10 km. The error bars represent the standard deviation of the measurements.

of 1.35 and 1.29, respectively. There might be several reasons for this, e.g. systematic uncertainties in the measurements and representativeness issues. More importantly, identical results are not expected given the different locations sampled by the two techniques. The OMI bias depends on the magnitude of the column, but likely also on other parameters. In absence of additional information, we adopt a bias-correction of OMI data based on both aircraft and FTIR data, shown as orange line

in Fig. 5(b). It is obtained by averaging the slopes and intercepts of the regressions of Fig. 2 and Eq. 3. The bias-corrected





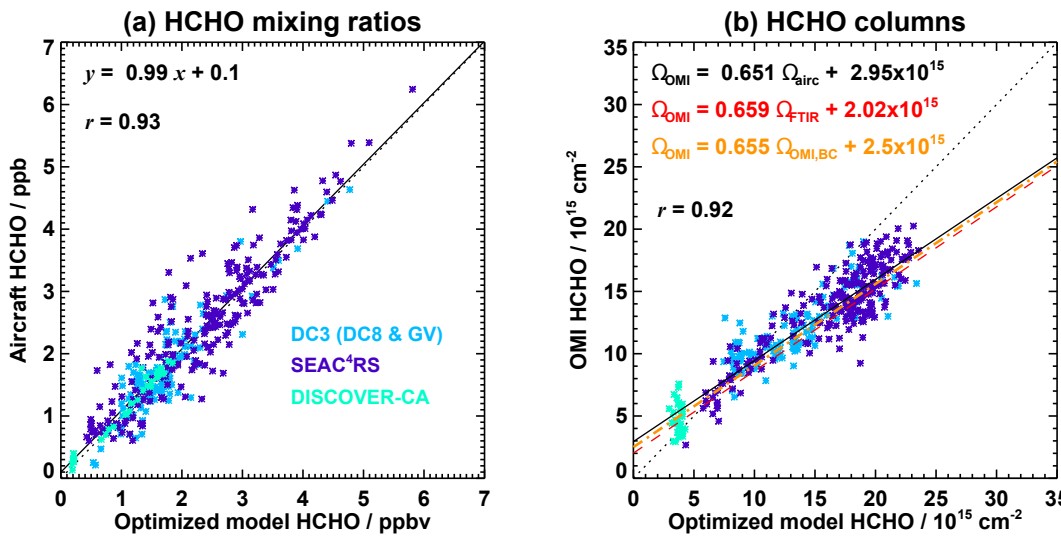

**Figure 5.** Scatter plots of (a) modelled and observed HCHO mixing ratios (daytime, below 4 km altitude) from three aircraft campaigns (DC3, SEAC⁴RS and DISCOVER-AQ California), and (b) modelled and OMI total HCHO columns at the same model pixels as in panel (a). The modelled values are constrained by the aircraft measurements through an emission optimization as described in text. The correlation coefficient and regression parameters using the Theil-Sen estimator are given in each panel (black font) and shown as black solid line. The slope and intercept of the regression of OMI columns vs. FTIR data (see Sect. 4.1) is given in panel (b) (red font) and shown as dashed red line. The adopted bias correction (relationship between OMI columns $\Omega_{OMI}$ and bias-corrected columns $\Omega_{OMI,BC}$) is also given (orange) and shown as orange dash-dotted line.

columns ($\Omega_{OMI,BC}$) are calculated with

$$\Omega_{OMI,BC} = (\Omega_{OMI} - 2.5 \cdot 10^{15})/0.655. \tag{4}$$

The resulting correction enhances columns above $\sim 7 \times 10^{15}\,\mathrm{molec.\,cm^{-2}}$ and decreases columns below that value. The contrasts between high- and low-emission regions will therefore be strengthened by the bias correction.

Note that the standard QA4ECV, cloud-corrected v1.2 product shows very similar biases with respect to aircraft data as the cloud-uncorrected product evaluated above. The evaluation of the standard product (CF$< 0.4$, with cloud correction) yields a slope of 0.68 and an intercept of $2.6 \times 10^{15}\,\mathrm{molec.\,cm^{-2}}$ (see Fig. S2).

## 5   Inferring emissions from bias-corrected OMI HCHO columns

### 5.1   Impact of bias correction on top-down VOC emissions

The global emission inversions are labeled as indicated in Table 2. OPT1 is conducted without any bias correction to the OMI HCHO columns, whereas OPT2 uses bias-corrected columns obtained from Eq. 4. OPT3 is a sensitivity test aimed at




**Table 2.** Global emission optimizations conducted in this work.

| Label | Description | Years |
|-------|-------------|-------|
| OPT1 | OMI-based inversion, without bias correction | 2006, 2008, 2011–2017 |
| OPT2 | OMI-based inversion, with bias correction | 2005–2017 |
| OPT3 | as OPT2, higher prior emission errors (factor 4) | 2012–2013 |

determining the influence of prior errors on the emission parameters. The errors on all emission parameters are taken to be a factor of $\sim 4\,(e^{1.4})$ in this case (instead of a factor of $e^{1.1} =\sim 3$ in OPT1 and OPT2, see Sect. 3.3).

Figure 6 shows seasonally-averaged HCHO columns from OMI (bias-corrected), the a priori simulation and the OPT2
simulation. Although the a priori simulation reproduces many features of the observations, there are noticeable differences, such as overestimated columns over Australia and Paraguay in December-January-February (DJF) and underestimated columns over southern Africa, Europe, East and South Asia in both DJF and June-July-August (JJA). The overestimation over Australia might be partly due to the high emission rates used in MEGANv2.1, based on measurements for young eucalypt trees which may emit more isoprene than adult trees (Emmerson et al., 2016).

As expected, the optimization realizes a much better agreement with the data, especially in areas with strong signal, such as Tropical regions during DJF and JJA, and Eastern U.S., China and India during summer. Model underestimations remain significant over regions with relatively low columns, such as Western U.S. and Europe, where the relative errors on the data are larger, particularly in winter; the retrieval error is typically $40-50\%$ over high-emission areas (e.g. Tropical forests) and $50-100\%$ over low-emission regions (e.g. Western Europe).

The improved simulation of HCHO columns from Fig. 6 is realized through significant changes in the amount and distribution of VOC emissions. Figure 7 displays the distribution of the emission ratios (optimized flux/a priori flux) per emission category, for both optimizations OPT1 and OPT2 (2011–2017 average). The mean annual emission totals in large regions are given in Table 3 for OPT2 (for 2005–2017), and in Table S3 for both OPT1 and OPT2 over 2011–2017.

The bias correction of OMI columns has a large impact on the inferred top-down emissions (Fig. S4), especially for the
biogenic emission category, which is the dominant contribution to HCHO columns over continental regions (Stavrakou et al., 2009). Without bias correction, global isoprene emissions are decreased from $430\,\mathrm{Tg\,yr^{-1}}$ in the a priori from MEGAN (2011–2017 period, Table S3) to $362\,\mathrm{Tg\,yr^{-1}}$ in optimization OPT1 (i.e. a $16\%$ decrease), whereas OPT2 increases those emissions by about $5\%$, to $451\,\mathrm{Tg\,yr^{-1}}$. The impact is strongest over South Asia ($+43\%$ difference between OPT1 and OPT2). It is also striking over many other regions, such as the Eastern U.S. and Southern China (Fig. 7), where biogenic emissions are decreased
by up to over $30\%$ in OPT1 but are increased in OPT2. In other regions such as the high latitudes and Australia, the difference is lower or even negative. This result is expected, given the lower columns in these regions (typically below $10^{16}\,\mathrm{molec.\,cm^{-2}}$), implying that the bias correction leads to only slight HCHO increases or even to HCHO decreases.





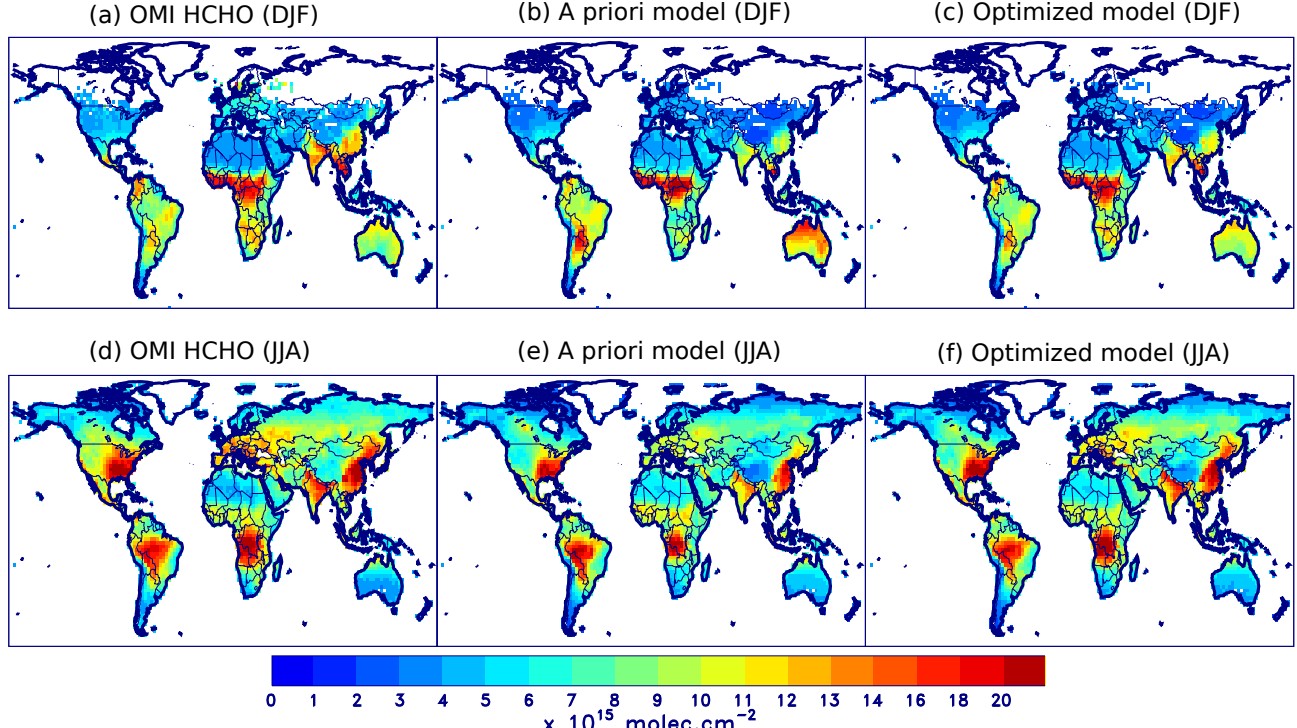

**Figure 6.** 2005-2017 average of HCHO columns ($10^{15}$ molec. cm$^{-2}$) in December-January-February (DJF) from (a) OMI (bias-corrected as described in the text), (b) the a priori model, and (c) the model with optimized emissions. (d)-(e), same as (a)-(c) but for June-July-August (JJA).

 

The top-down global isoprene emissions from OPT2 (445 Tg yr$^{-1}$ for 2005–2013) are a factor of 1.64 higher than the top-down emissions derived by inverse modelling of OMI data by Bauwens et al. (2016) (272 Tg yr$^{-1}$). Note that our a priori (MEGAN) emissions are also higher than in Bauwens et al. (2016), especially over Australia and Africa, due to the soil moisture stress impact that was accounted for by Bauwens et al. (2016) but not in our study (Sect. 3.2). The bias correction of OMI data applied in this work explains only a part (factor 1.25) of the large discrepancy between the top-down estimates. The main reason for the remainder is the different OMI retrieval (BIRA-V14) (De Smedt et al., 2015) used as constraint in Bauwens et al. (2016). The BIRA-V14 HCHO columns are indeed significantly lower than the QA4ECV data used here, as already noted by Wells et al. (2020). As shown on Fig. S3, QA4ECV columns are typically higher than the V14 product by $10-50\%$ over continents, and by up to $80\%$ in parts of Southern Africa. This explains why the OPT1 isoprene emissions over Southern Hemisphere Africa, $60$ Tg yr$^{-1}$, are about a factor of 2 higher than the top-down estimate from Bauwens et al. (2016), whereas the two top-down estimates are in much better agreement over South America and Australia, where the QA4ECV and V14 HCHO columns are more similar (Fig. S3).




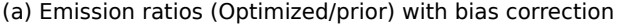

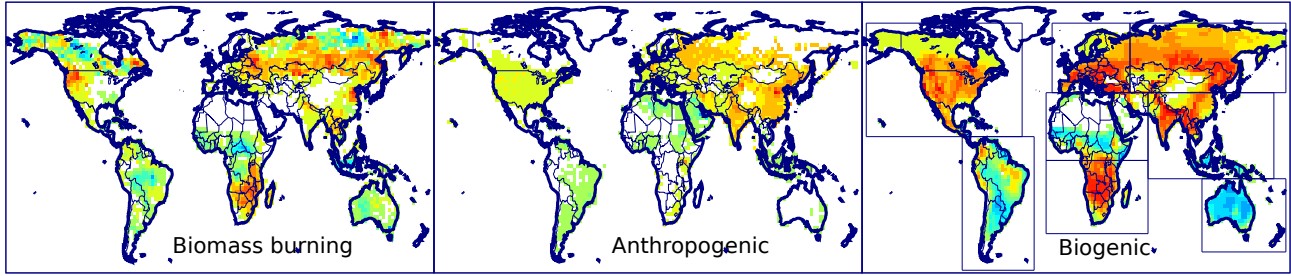

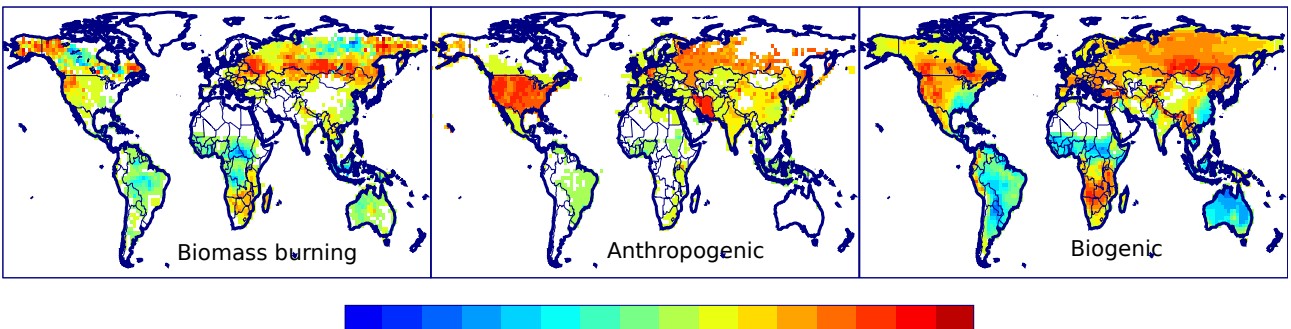

**Figure 7.** Ratios of top-down emissions to a priori emissions (2011–2017 average) for biomass burning VOCs (left panel), anthropogenic VOCs (middle), and biogenic VOCs (right), for optimizations (a) OPT2, constrained by bias-corrected columns, and (b) OPT1, constrained by uncorrected columns. Pixels with very small emission changes ($< 1\%$) are left blank. The regions used for calculation of total emissions given in Table 3 are shown as boxes on the top right panel.

The effect of the bias correction on the optimized emissions is obtained by comparing the OPT1 and OPT2 top-down emissions. The bias correction increases the top-down biogenic emissions (by a factor of 1.25 at global scale) and biomass burning emissions (factor of 1.13), and decreases the anthropogenic emissions at mid- and high latitudes, except over China (Table S3 and Fig. 7). This is primarily due to the generally low HCHO columns during winter in those regions, typically below $6.5 \times 10^{15}\,\mathrm{molec.\,cm^{-2}}$, for which the bias correction decreases the columns (Eq. 4). The exacerbated seasonal variation of the HCHO columns due to the bias correction favors the biogenic emissions to the detriment of anthropogenic sources at these latitudes. This result is highly dependent on the HCHO column uncertainties, however. The scatter plots of Fig. 2 and Fig. 5 shows a large dispersion for low columns, indicating high uncertainty in the bias correction. For this reason, the optimization results should be considered with caution in low-column areas.





**Table 3.** Mean a priori and OMI-based (OPT2) emission estimates ($\mathrm{Tg\,yr^{-1}}$) per source category for different world regions and globally. Regions are defined in Fig. 7. The means are taken either over 2005-2017 (OPT2 simulation period) or 2005-2013 (for comparison with Bauwens et al. (2016)). B16: Bauwens et al. (2016). NH: Northern Hemisphere; SH: Southern Hemisphere.

| | North America | South America | Europe | NH Africa | SH Africa | North Asia | South Asia | Oceania | **Global** |
|---|---|---|---|---|---|---|---|---|---|
| Biomass burning NMVOC emissions | | | | | | | | | |
| GFED4s 2005–2017 | 5.9 | 15.0 | 1.7 | 16.3 | 25.6 | 6.8 | 14.4 | 3.9 | **90** |
| OPT2 2005–2017 (this work) | 5.4 | 12.5 | 2.2 | 12.7 | 30.2 | 8.2 | 12.4 | 3.7 | **87** |
| Isoprene emissions | | | | | | | | | |
| MEGAN-MOHYCAN 2005–2017 | 35 | 149 | 6.7 | 85 | 44 | 9.3 | 39 | 60 | **430** |
| OPT2 2005–2017 (this work) | 44 | 129 | 11.4 | 77 | 76 | 15 | 53 | 36 | **443** |
| OPT2 2005–2013 (this work) | 45 | 129 | 11.2 | 78 | 76 | 14 | 51 | 36 | **445** |
| MEGAN-MOHYCAN 2005-2013 (B16) | 32 | 141 | 6.8 | 50 | 29 | 9.4 | 36 | 38 | **343** |
| OMI-based 2005–2013 (B16) | 26 | 97 | 8.4 | 35 | 28 | 11 | 31 | 36 | **272** |
| Anthropogenic NMVOC emissions | | | | | | | | | |
| EDGAR 2005–2017 | 20.6 | 12.6 | 17.0 | 39.4 | 11.8 | 11.3 | 52.2 | 1.1 | **166** |
| OPT2 2005–2017 (this work) | 21.4 | 11.5 | 18.8 | 34.8 | 12.8 | 14.7 | 59.8 | 1.1 | **175** |
| **Total NMVOC emissions** | | | | | | | | | |
| A priori 2005–2017 | 62 | 176 | 25 | 141 | 82 | 28 | 106 | 65 | **696** |
| OPT2 2005–2017 (this work) | 71 | 154 | 32 | 128 | 120 | 38 | 126 | 43 | **730** |

## 5.2 Evaluation of optimized HCHO against aircraft and FTIR data

The optimizations OPT1 and OPT2 were conducted over years (Table 2) for which aircraft campaign data are available, over the U.S., Canada, Mexico and South Korea (Fig. 1 and Table 1). Figure 8 displays the mean observed and modelled vertical HCHO profiles for all campaigns. For most campaigns, the model agreement is improved when the bias correction is applied to the satellite columns. On average for all campaigns, the large negative bias found for both the a priori simulation ($-27\%$) and the OPT1 run ($-28\%$) is strongly reduced in the OPT2 run ($-9\%$) (Table S4). The root mean square deviation (RMSD)

is also decreased, from 0.85 and 0.86 ppbv in the a priori and OPT1 run to 0.55 ppbv in the OPT2 run. In particular, OPT2 presents very little biases over high-column areas (e.g. DC3, SEAC[4]RS, SENEX). The observed horizontal distribution of the vertically-averaged HCHO mixing ratios is also very well matched by the model over high-emission regions in the Eastern U.S., with Pearson's correlation coefficients ranging between 0.96 and 0.98 after optimization for those campaigns (Fig. S5).

In contrast with this good performance, OPT2 presents large underestimations over low-emission areas over the Western

U.S., namely the DISCOVER campaigns in Colorado ($-43\%$ bias) and California ($-53\%$). Low OMI columns were measured during the latter campaign (Jan.–Feb. 2013), in the range $(3-6)\times10^{15}\,\mathrm{molec.\,cm^{-2}}$, such that the bias correction further





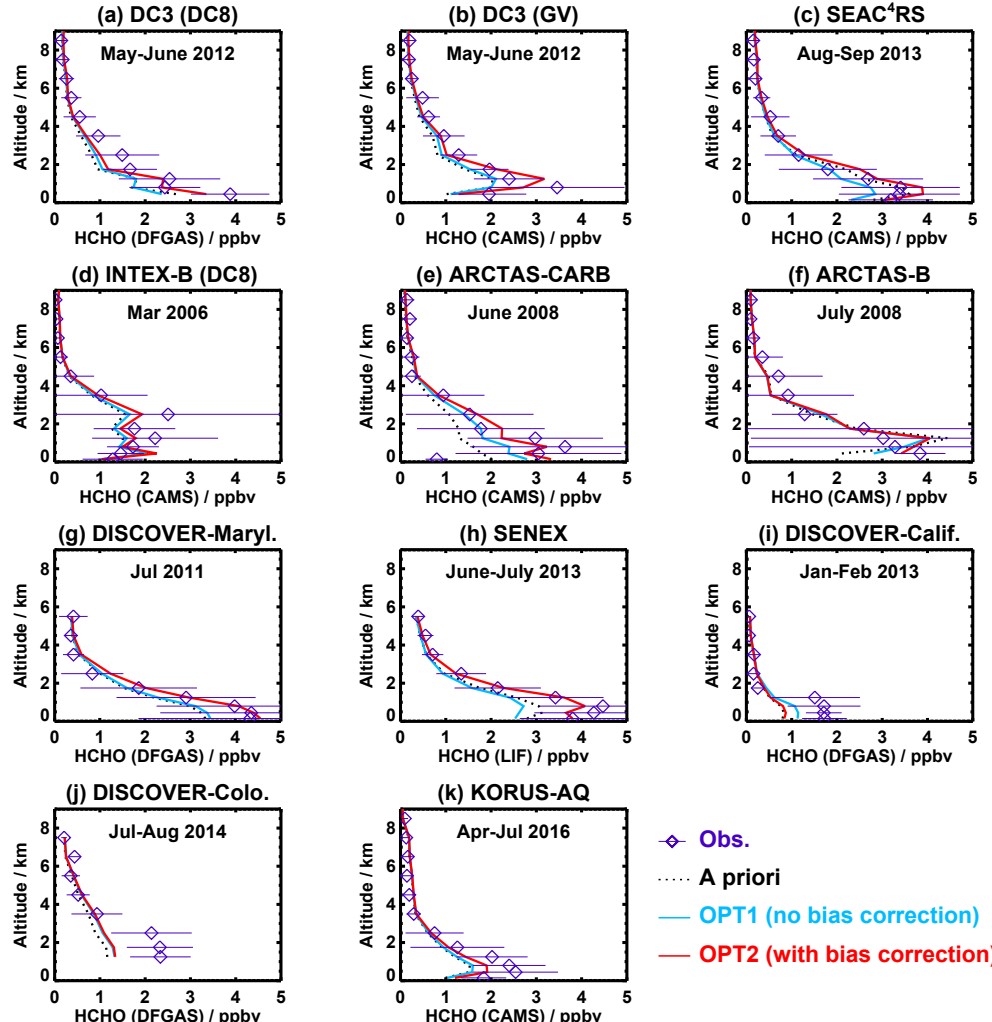

**Figure 8.** Measured and modelled profiles of HCHO mixing ratios over land (in ppbv) for (a) DC3 (DC8), (b) DC3 (GV), (c) SEAC$^4$RS, (d) MILAGRO, (e) ARCTAS-CARB, (f) ARCTAS-B, (g) DISCOVER-AQ Maryland, (h) SENEX, (i) DISCOVER-AQ California, (j) DISCOVER-AQ Colorado, and (k) KORUS-AQ. Optimization constrained by either uncorrected OMI columns (OPT1, in blue) or bias-corrected columns (OPT2, in red). The error bars represent the standard deviation of the measurements. Comparison statistics are provided in Table S4.

decreased the columns. The OMI columns were higher during DISCOVER-Colorado ($\sim 7 \times 10^{15}$ molec. cm$^{-2}$ near Boulder, Colorado in July–Aug. 2014), but the modelled columns remained much underestimated (by $0 - 40\%$) after emission optimization, due to the large OMI column uncertainties.

Figure 9 displays the mean observed and modelled seasonal evolution of HCHO columns at 11 FTIR stations located in source regions. The other stations (primarily high-altitude, high-latitude or maritime stations) show little sensitivity to VOC



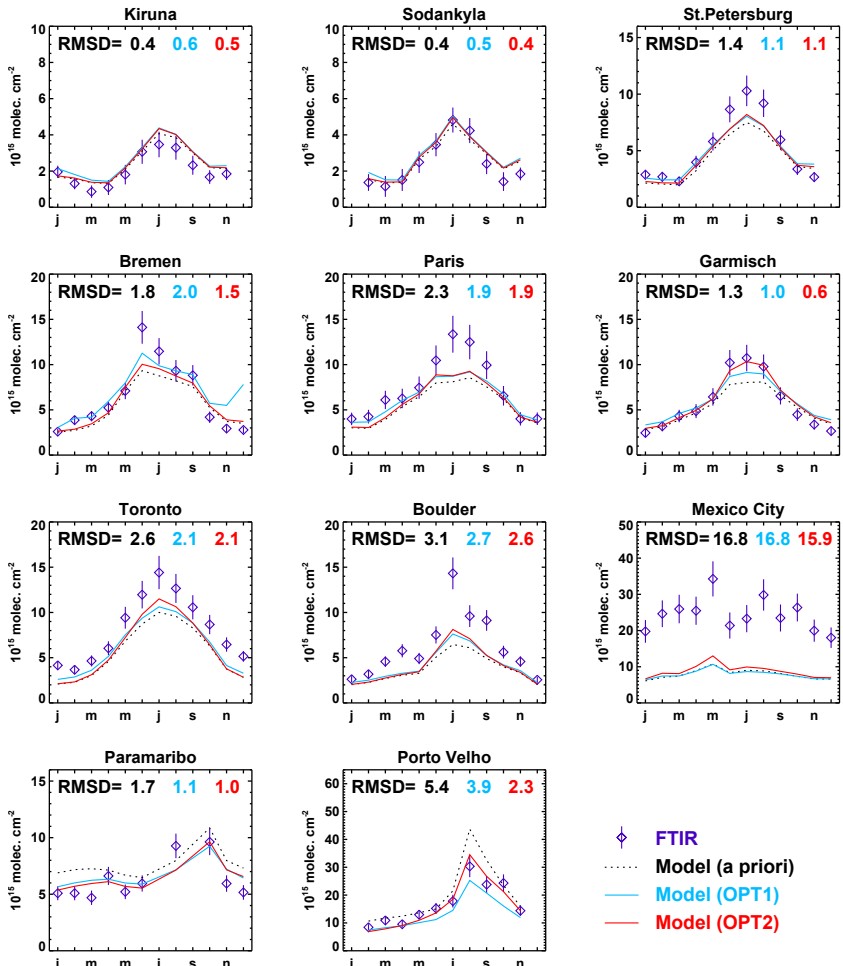

**Figure 9.** Average seasonal cycle of HCHO columns at FTIR stations (2011-2017). Symbols with error bars represent the mean FTIR columns and related systematic uncertainties. Black dotted curve: model with a priori emissions; blue curve: OPT1 simulation constrained by uncorrected OMI columns; red curve: OPT2 simulation constrained by bias-corrected OMI columns. The root mean square deviation (RMSD) of the 3 model runs is given inset for each station.

emission changes. The complete time series (2005–2017) of monthly observed and modelled (OPT2) HCHO columns are shown in the Supplement (Figs. S6 and S7). The impact of the emission optimizations is relatively small at mid- and high latitudes, and largest at the two South American stations (Paramaribo and Porto Velho). The OPT2 run performs better than OPT1, with lower RMSD values at all sites and lower biases at 8 out of 11 sites (Table S5). The OPT2 run also performs better than the prior simulation at all sites except Kiruna and Sodankyla in northern Scandinavia, where the moderate biogenic emission increase inferred by the optimization (Fig. 7) enhances the small positive model bias which was already present in the a priori simulation.






At the other European stations and at North American sites, the emission increase of the OPT2 run improves the agreement
with the measurements, but fails to close the gap completely (except at Garmisch), in particular during summer. This discrepancy is mostly explained by the model underestimation of OMI columns after optimization, as seen in Fig. 6 and Fig. S8.
The exception is Mexico City, where the model successfully matches the OMI columns (Fig. S8) but underestimates the FTIR
columns by about a factor of 3. This poor model performance can be attributed to the coarse model resolution ($2° \times 2.5°$) being
unable to resolve the megacity emissions. At Boulder, the model underestimation of both OMI and FTIR columns (by up to a
factor of 2 in summer) is consistent with the model underestimation against airborne HCHO data from DISCOVER-Colorado,
by almost a factor of 2 (Fig. 8).

The comparison at the Amazonian sites of Paramaribo (Suriname) and Porto Velho (Rondônia, Southwestern Brazil) validates the VOC emission decrease (by $\sim 20\%$) inferred by OPT2 during the dry season (July–November). The stronger emission
decrease of the OPT1 run in Rondônia (factor of 2) leads to a significant model underestimation of FTIR HCHO columns at
Porto Velho. At both sites, biogenic emissions are also strongly decreased in the first half of the year (up to a factor of 3 at
Porto Velho in February 2017). This decrease at Porto Velho is due to very low HCHO columns observed by OMI in the first
months of 2017, especially February (Fig. S8).

### 5.3 Evaluation of top-down isoprene emissions

The major source regions of biogenic isoprene according the MEGAN model are South America, Sub-Saharan Africa, Australia, the Eastern U.S. and South Asia, including South China (Fig. 10a). The fluxes display a pronounced seasonality reflecting
primarily their dependence on temperature and radiation (Guenther et al., 2006), with clear maxima during summertime, except
near the Equator. In January 2013, the highest fluxes are predicted in a region centered over Paraguay in South America as well
as over northern and eastern Australia; in April, near the Peru-Brazil border region and especially over the Central African
Republic (CAR) and South Sudan; and in July, in the U.S. states bordering the Mississippi River below $38°N$ latitude.

The OMI-based optimization (OPT2) leads to substantial changes in the distribution of isoprene emissions. As seen on
Fig. 10, sharp declines are predicted over most hotspots mentioned above, most noticeably those over Paraguay, the CAR,
Australia and the Peru-Brazil border. Emission decreases are also found over the Guayanas and Indonesia, especially in January.
Other regions emerge as important source regions, most prominently the Congo Basin and a vast area in southwestern Africa
spanning Angola, Namibia and Botswana. Although the distributions shown on Fig. 10 are for the year 2013, similar features
and emission updates are derived for all years (see Sect. 5.4 for a discussion of interannual variability).

Those emission updates are in good qualitative agreement with optimized isoprene emissions based on isoprene column
densities from the spaceborne Cross-track Infrared Sounder (CrIS) for the same year (2013) (Wells et al., 2020). Those authors applied an iterative mass balance technique to optimize the emissions of isoprene in the GEOS-Chem model, using
as constraints the first global retrieval of isoprene columns from space. Due to the critical role played by hydroxyl radical
(OH) levels in this emission inversion, and given the strong influence of nitrogen oxides ($NO_x$) on OH, the $NO_x$ emissions
in GEOS-Chem were optimized based on spaceborne $NO_2$ data before conducting the isoprene inversion. The CrIS-derived
monthly-averaged emissions (Supplementary Figs. 12–15 in Wells et al., 2020) display striking similarities and differences





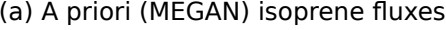

(a) A priori (MEGAN) isoprene fluxes

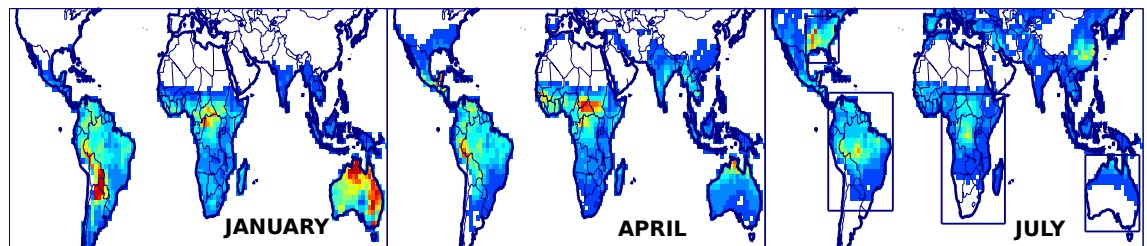

(b) OMI-based (bias-corrected) isoprene fluxes

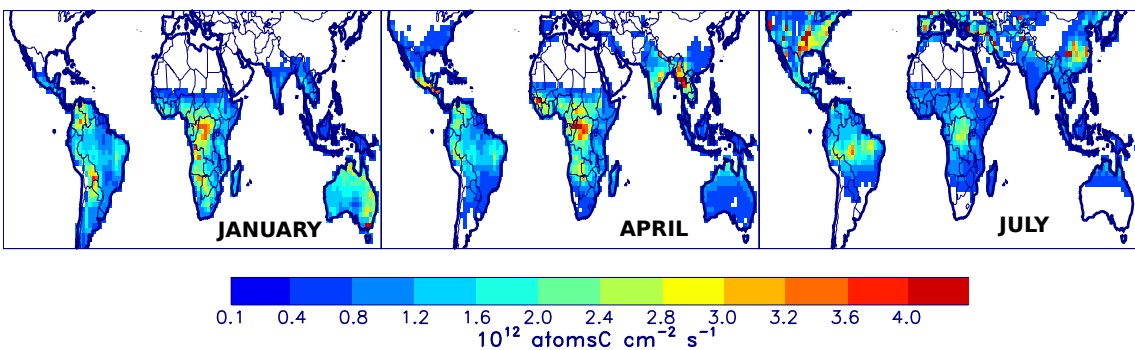

**Figure 10.** (a) A priori (MEGAN) isoprene monthly fluxes ($10^{12}$ atoms C cm$^{-2}$ s$^{-1}$) in year 2013. (b) Optimized isoprene fluxes based on (bias-corrected) OMI HCHO columns, also for 2013. The boxes shown in the July a priori flux subplot show the regions for which emission totals are displayed in Fig. 11.

with the OMI-constrained distributions of Fig. 10b. Common features include the decreases over Paraguay, the Guyanas, the Peru-Brazil border, and the CAR and South Sudan region, as well as the increases over Congo and southwestern Africa as well as Colombia and Central U.S. states spanning from Texas to Illinois. Despite very different a priori isoprene emissions over Australia (e.g. respectively 10.8 and 4.2 Tg in January in our inventory and in Wells et al., 2020), the CrIS-based total emissions over the continent are slightly higher than our OMI-based values (Fig. 11). Closer examination shows a good agreement in the Eastern part, whereas the CrIS-based emissions appear to be higher in the North and lower in the Central part of the country, compared to our estimates (Fig. 10). The OMI-based emissions are significantly lower than both the a priori estimate and the CrIS-based fluxes over South America in January and April, whereas the opposite holds over Africa (Fig. 11). The discrepancy is largest over Brazil, where OMI columns as low as $6 \times 10^{15}$ molec. cm$^{-2}$ are frequently observed during the wet season (December–May). Those low values drive a decrease in biogenic emissions, reaching ca. $20 - 30\%$ over Western and Northern Brazil. As discussed in Sect. 5.2, the optimized model underestimates FTIR data in February–May at Porto Velho, due to excessively low OMI columns (in February). Although the Porto Velho comparison was for a different year (2017), it suggests that the wet season OMI data might be less reliable, possibly due to high cloudiness in the area. In July, a very good agreement is found between the CrIS-based and OMI-based emissions over South America, in terms of distribution and total.





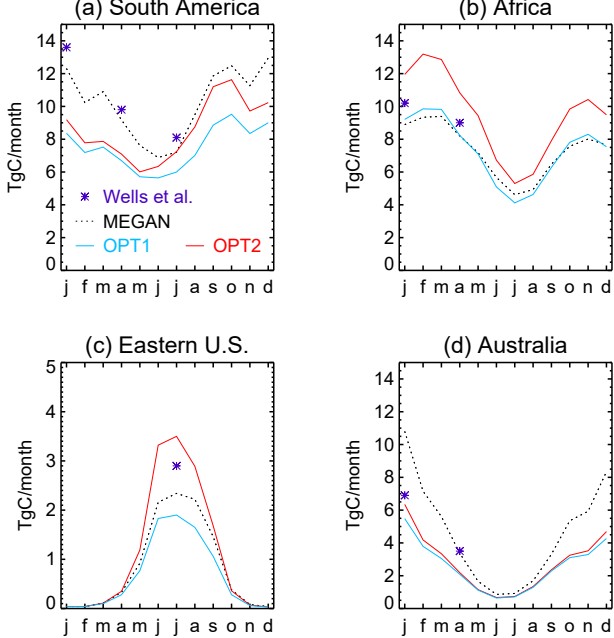

**Figure 11.** Seasonal variation of isoprene monthly fluxes (2013) in (a) South America (82.5–32.5 W, 32 S–14 N), (b) Africa (5–52.5 E, 36 S–14 N), (c) Southeastern U.S. (75-100 W, 26–42 N), (d) Australia (115–155 E, 10–40 S). Results shown for the a priori inventory (MEGAN, dotted line), the OMI-based optimizations (OPT1 in blue, OPT2 in red) and the CrIS-based optimization results from Wells et al. (2020) (asterisks).

Over Africa, the discrepancy is largest in January over the CAR, in a region with strong fire emissions in several biomass burning inventories including GFEDs (Pan et al., 2020). Both pyrogenic and biogenic VOC emissions are strongly reduced in this region by the OMI-based inversion, as in several previous inverse modeling studies (e.g., Bauwens et al., 2016; Müller et

al., 2018), but the interference of strong pyrogenic emissions makes the top-down isoprene estimate particularly uncertain, and likely contributes to the differences with CrIS-based results. Interference from biomass burning probably plays a role in other regions as well, especially Southern Africa in June–September, the southern part of Amazonia in August–September, southeast Asia in February–April, and boreal forests during June–August. In most other regions and periods, the discrimination between pyrogenic and biogenic emissions is less critical, given the general dominance of biogenic VOCs over biomass burning as

source of HCHO (Stavrakou et al., 2009, 2018). In addition, the derivation of isoprene emissions from CrIS data has also its limitation, most importantly the role played by the assumed isoprene vertical profile in the retrieval of the total column (Wells et al., 2022) and the strong impact of OH radical concentrations on CrIS-derived emissions (Wells et al., 2020).





**Figure 12.** Annually-averaged (bias-corrected) HCHO columns (2005–2017) averaged over (a) Amazonia (40–75° W, 20° S–5° N), (b) S-E U.S. (75–100° W, 26–36° N), (c) Southern Africa (20° W–65° E, 0–40° S), (d) Australia (110–155° E, 10–38° S), (e) India (67–92° E, 10–30° N), (f) Middle East (40–55° E, 26–42° N), (g) North China plain (112–122° E, 32–40° N), (h) N-E China (122–132° E, 42–50° N), (i) W Canada (100–120° W, 54–64° N). Units: $10^{15}\,\mathrm{molec.\,cm^{-2}}$. Linear regression trends (2005–2016) are given inset. OMI: violet symbols; a priori model in blue, optimized model (OPT2) in red.

## 5.4   Trends of HCHO columns and VOC emissions

Figure 12 displays the temporal evolution of annually-averaged (bias-corrected) OMI and modelled columns over large regions.
The contours of those regions are shown on Fig. 13, which displays the distribution of percentage trends of HCHO columns (panels a-c) and total VOC emissions (d-e) between 2005 and 2016.



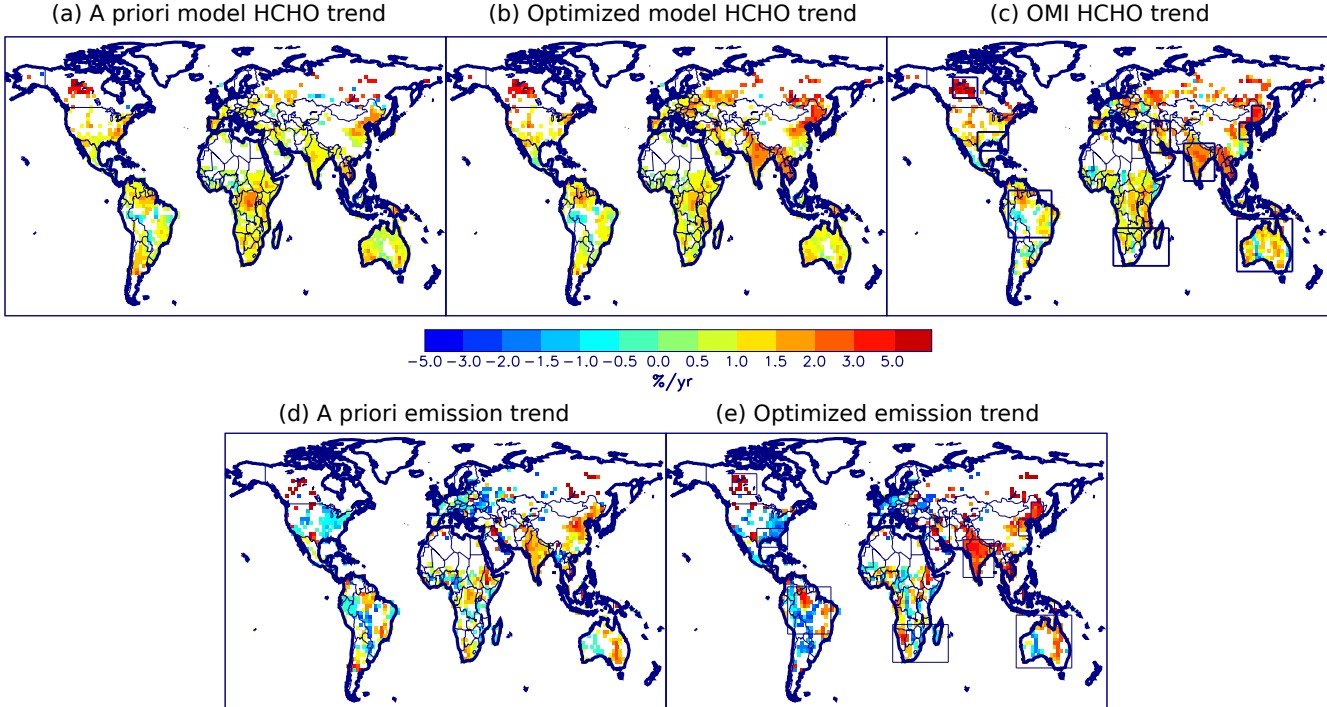

**Figure 13.** Trends (%/yr) over 2005–2016 of yearly-averaged HCHO columns from (a) the a priori model simulation, (b) the optimized (OPT2) simulation, (c) the bias-corrected OMI retrieval, and trends of total NMVOC emissions from (d) the a priori simulation, and (e) the OPT2 simulation. Pixels with low a priori emission (below $5 \times 10^{10}\,\mathrm{molec.\,cm^{-2}\,s^{-1}}$) or for which the trend uncertainty exceeds 150% of the trend are left as blank. The boxes in panels (c) and (e) show the regions used for calculating the temporal evolution of averaged columns (Fig. 12) and total emissions (Fig. 14).

Already without emission optimization, the modelled columns are strongly correlated temporally with the observations over regions where biogenic VOC emissions and biomass burning are the dominant sources of HCHO. Interannual variability over those regions is primarily related to climate variables (temperature and radiation) through their influence on biogenic emissions,
biomass burning and background HCHO (Stavrakou et al., 2018), and the good model performance suggests that the inventories used in the model (MEGAN-MOHYCAN and GFED4s) are generally adequate. The year 2017 stands out in the time series over several regions in the Southern Hemisphere, namely Amazonia, Southern Africa and Australia (Fig. 12). The temporal correlation between OMI and the a priori model deteriorates when including year 2017, and a further degradation is seen when also including 2018 (not shown). This degradation is found not only for those large regions but also on maps of the correlation
coefficient for the different time periods (not shown). The average correlation coefficient over the Southern Hemisphere (over $2° \times 2.5°$ pixels with mean columns higher than $5 \times 10^{15}\,\mathrm{molec.\,cm^{-2}}$) is decreased from 0.62 for 2005–2016 to 0.57 for 2005–2017 and 0.56 for 2005–2018. This deterioration is likely related to instrumental degradation and/or to changes in the





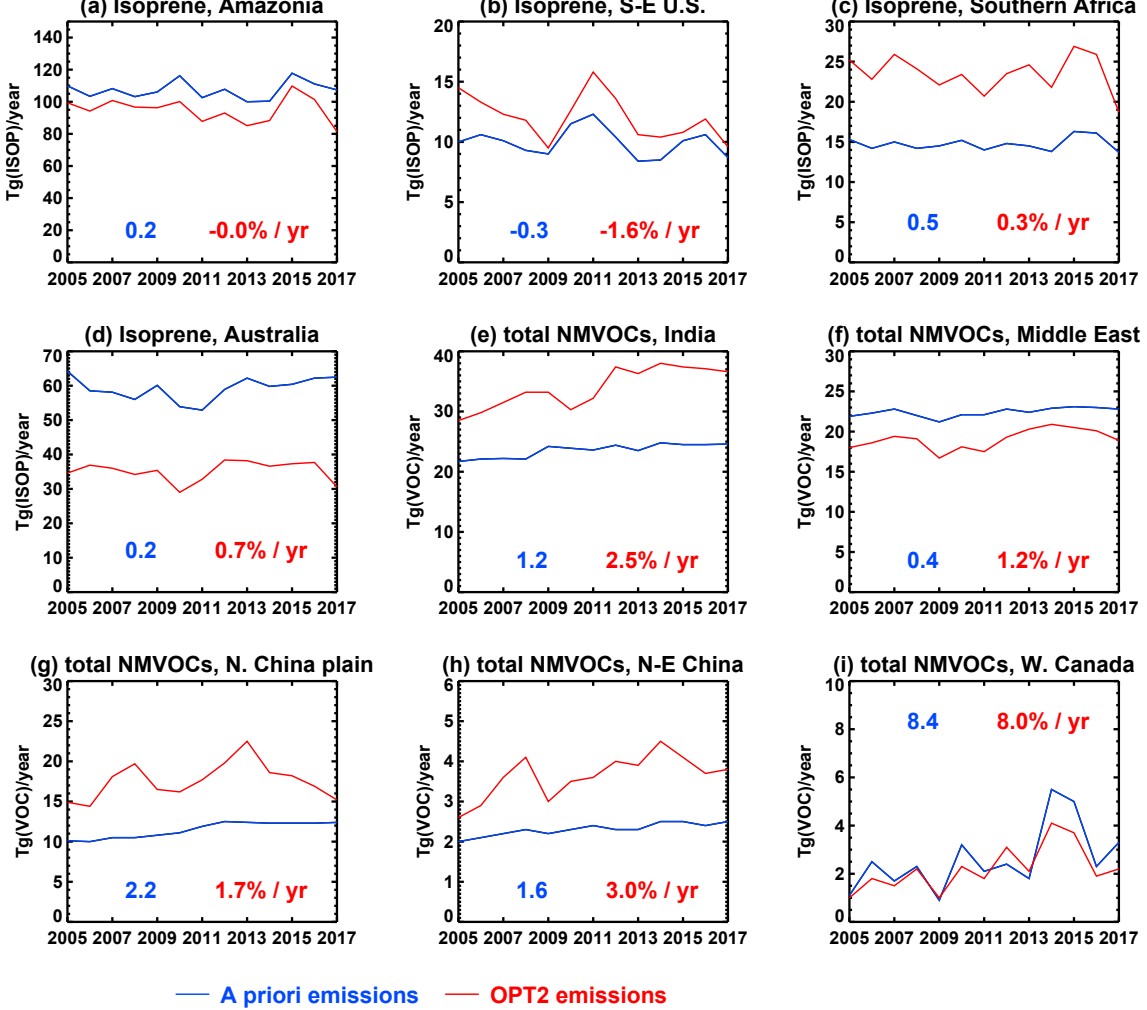

**Figure 14.** (a-d) Total isoprene emissions, (e-h) total VOC annual emissions $(\mathrm{Tg\,yr^{-1}})$ from the same large regions as in Fig. 12. A priori emissions in blue, optimized emissions (OPT2) in red. Linear regression trends (2005–2016) are given inset.

version of the model (TM5) used for estimating the air mass factors. We therefore restrict the remaining discussion to the period 2005–2016.

The observed and modelled trends of yearly HCHO columns are positive over most areas (Fig. 12 and Fig. 13a-c). Those trends in HCHO are partly explained by emission trends (Fig. 13d-e and Fig. 14), e.g. the rapid increase of anthropogenic VOC emissions over many Asian countries $(> 1\%\,\mathrm{yr^{-1}}$ over e.g. India and China), the rapid intensification of boreal forest fires over Western Canada and Eastern Siberia (up to $\sim 4\%\,\mathrm{yr^{-1}}$), and comparatively slower increases $(< 1\%\,\mathrm{yr^{-1}})$ in biogenic VOC emissions over many areas due to global warming (e.g. over Amazonia, Southern Africa and Australia) (Fig. 14).



The trends derived by the a priori model in regions with strong anthropogenic influence (India, Middle East, northern China) are underestimated, partly because the a priori anthropogenic emissions are kept constant in the model after 2012 and taken equal to their 2012 values (from EDGAR). The largest discrepancy between the model and OMI is seen over India and Northeastern China (North of $42°$N, see Fig. 13c), where the observed trends are about twice larger as in the model. As a result, the top-down emission trends in those regions are among the highest globally, reaching $2.5 - 3\%$ per year over 2005–

2016 (Fig. 14). Over the last years, however, the emissions appear to level off over India (since 2012) and even to decrease over the North China Plain (since 2013), likely due to emission regulations (Bauwens et al., 2022).

Over the Eastern U.S., both the a priori and optimized emission trends are generally negative, in spite of an increasing trend in yearly HCHO columns (Fig. 13). This apparent paradox is due to a strong seasonality in trends, as the summertime (May–September) HCHO trends are much lower (more negative) than the annual trends (Fig. S9–S10). For example, the summertime

HCHO OMI trend over the Southeastern U.S. is $-0.4\%\,\mathrm{yr}^{-1}$, i.e. $0.7\%\,\mathrm{yr}^{-1}$ below the annual trend; a similar difference is found in the model results. The significant decreasing trend of summertime OMI column over this region, already noted in previous studies (De Smedt et al., 2015; Zhu et al., 2017; Opacka et al., 2021), is difficult to explain. Zhu et al. (2017) proposed that this trend is partly due to the decline in anthropogenic $NO_x$ emissions caused by air quality regulations. Indeed, the yield of HCHO from VOC oxidation is believed to decrease when $NO_x$ levels decrease. However, this effect was found to account for

only $\sim 20\%$ of the HCHO column change seen by OMI, based on GEOS-Chem model calculations. The OMI HCHO decline is therefore more likely due to changes in emissions. Zhu et al. (2017) noted that the strong HCHO decline over the Houston-Galveston-Brazoria area likely reflects a fast decrease in anthropogenic point source VOC emissions, but the spatial extent of this effect should be limited, whereas negative trends in OMI columns (and in top-down VOC emissions) are derived over a much vaster area. This suggests a trend in the emissions of isoprene, as its oxidation is believed to be the dominant source

of HCHO over the Southeastern U.S. (Palmer et al., 2006). Possible drivers of such trend (besides temperature and visible radiation fluxes, already considered in the MEGAN model) include soil moisture stress, $CO_2$ inhibition and land-use change. The effect of soil moisture stress is ignored in this study, given the difficulties associated to its parameterization (Bauwens et al., 2016; Opacka et al., 2021), but it could be significant in this area. The effect of $CO_2$ is considered in our a priori emissions using the parameterization of Possell and Hewitt (2011), which induces a emission decline of about $\sim 0.45\%\,\mathrm{yr}^{-1}$

over 2005–2016, but this effect is highly uncertain.

Using high-resolution tree cover information from the Global Forest Watch database (Hansen et al., 2013), Opacka et al. (2021) showed that a widespread decline in tree cover occurred in the Eastern U.S. over 2001–2016. The estimated trends in isoprene emissions due to land-use change between 2005 and 2016 at the model resolution ($2° \times 2.5°$), calculated according to Opacka et al. (2021), are shown on Fig. S11. The tree cover decline induces negative trends in isoprene fluxes that are generally

small ($< 0.5\%\,\mathrm{yr}^{-1}$) but which might contribute to the large negative emission trend inferred from OMI data. Note that those changes due to land-use may be underestimated as they do not consider changes in species composition, e.g. the replacement of high isoprene emitters by comparatively lower isoprene emitters.

Land use change might impact other regions, especially South America. Extensive tree cover loss over a large region extending from Paraguay, Northern Argentina and Eastern Bolivia to the Brazilian States of Rondônia, Mato Grosso and Pará





(Opacka et al., 2021) has caused negative trends in isoprene emissions of up to $-2\%\,\mathrm{yr}^{-1}$ (Fig. S11), which likely contribute to the patterns of negative trends inferred from OMI over this continent (Fig. 13e). A more thorough analysis at higher spatial resolution would be needed to separate those effects from other possible drivers of changes in OMI columns (e.g. biomass burning), in South America as well as on other continents.

## 6    Conclusions

The HCHO vertical columns of the QA4ECV OMI dataset are evaluated using two qualitatively very different measurement types: ground-based vertical columns of formaldehyde measured at 26 FTIR stations worldwide, and in situ HCHO measurements from aircraft campaigns conducted over the U.S. in 2012–2013. A regional atmospheric chemistry-transport model is used to assimilate the airborne measurements and to derive HCHO distributions that can be compared to the satellite data while closely approximating the vertical and horizontal distribution of the measurements. The two datasets are complemen-

tary: whereas the FTIR dataset covers a wide range of conditions and is geographically more diverse, the aircraft campaigns used in this work cover a known major VOC source region and are particularly relevant to the evaluation of BVOC emissions using satellite data.

The evaluation of OMI columns against FTIR column data and airborne in situ measurements leads to similar conclusions, in spite of the large qualitative difference between the datasets. The regression parameters are remarkably similar, with nearly

identical slopes (0.66) and similar intercepts. On one hand, the large OMI columns (especially above $\sim 12\times10^{15}\,\mathrm{molec.\,cm}^{-2}$) are generally underestimated, by a factor of 1.29 (against aircraft data) to 1.36 (against FTIR) for an OMI column of $20\times10^{15}\,\mathrm{molec.\,cm}^{-2}$. On the other hand, low OMI columns (especially below $\sim 6\times10^{15}\,\mathrm{molec.\,cm}^{-2}$) are often underestimated, by $-10\%$ and $-37\%$ against FTIR- and aircraft-based regressions, respectively, for an OMI column of $5\times10^{15}\,\mathrm{molec.\,cm}^{-2}$.

The correlation of FTIR vs. co-located OMI data is relatively poor (Pearson's coefficient of 0.67), but the regression is

qualitatively consistent with the (much better-correlated) evaluation of TROPOMI HCHO column measurements against FTIR data (Vigouroux et al., 2020), although the latter suggests a larger underestimation of satellite columns with respect to FTIR data: a factor of 1.49 for a column of $20\times10^{15}\,\mathrm{molec.\,cm}^{-2}$ according to the regression of monthly averages (Vigouroux et al., 2020). To summarize, both aircraft and FTIR data suggest a significant underestimation (overestimation) of satellite data for high (low) columns, but the discrepancy is slightly lower according to aircraft data. Clearly, more work will be needed to refine

those estimates, and especially to elucidate and potentially eliminate the possible causes for disagreement. Ideally, a co-located cross-evaluation of FTIR columns and aircraft profiles (e.g. spiral flights) would be needed to assess their mutual consistency.

The OMI bias against FTIR and aircraft data is tentatively corrected through a linear relationship, and the bias-corrected HCHO columns are used as constraints to optimize the emissions of VOCs in the MAGRITTE global model over 2005–2017. Evaluation of the modelled distributions of HCHO against 11 aircraft datasets spanning 2006–2016 show that the bias cor-

rection leads to a considerable reduction of the average bias against aircraft observations, from $-27\%$ in the prior simulation and $-28\%$ in an optimization without bias correction to $-9\%$ when using bias-corrected data. The root mean square deviation (RMSD) is also decreased. Agreement with FTIR data is also improved, especially at South American sites, but the improve-



ment is marginal at mid-latitudes, where the relative uncertainties of the OMI columns are high and the bias correction has a very small effect on the OMI columns.

The optimized VOC emissions of the optimization using bias-corrected data (OPT2) are only slightly higher ($+5\%$ globally) than the a priori emissions, but are much higher than previous top-down estimates based on OMI data. In particular, the global top-down isoprene emissions of the OPT2 run ($445\,\mathrm{Tg\,yr}^{-1}$) are $64\%$ higher than those derived by Bauwens et al. (2016) for 2005–2013. Regionally (e.g. over Africa), the difference exceeds a factor of 2. The major reason for the increased top-down emissions is the much higher HCHO columns used in this work, due to the bias correction and the higher QA4ECV HCHO
columns, compared to the previous BIRA-V14 retrieval used by Bauwens et al. (2016). This result demonstrates the importance of validation for any reliable quantitative use of satellite HCHO data.

The biogenic emission updates of the OPT2 inversion bring the model closer to top-down isoprene emissions derived from the spaceborne (CrIS) isoprene columns (Wells et al., 2020) in many regions, including Central and Southern Africa, the Eastern U.S. and parts of South America. The comparison also highlights important differences, which are partly attributed to
known weaknesses of the HCHO-based inversion, such as the high uncertainties of low HCHO columns (e.g. over Europe and over Amazonia during the wet season) and the co-occurrence of very extensive vegetation fires (e.g. over the Central African Republic in the dry season). Furthermore, spaceborne isoprene data have their own uncertainties, evidenced by the substantial impact of recent algorithmic advances on the column magnitudes (Wells et al., 2022); in addition, the top-down emissions from CrIS are very dependent on OH radical levels, which remain uncertain in remote areas e.g. due to their sensitivity to $\mathrm{NO_x}$
(Wells et al., 2020) and to mechanistic uncertainties (Müller et al., 2019; Berndt et al., 2019; Novelli et al., 2020).

The interannual variability of HCHO columns is generally well explained over regions where biogenic emissions and biomass burning are the dominant source of VOCs. Large positive emission trends of up to $2-3\%\,\mathrm{yr}^{-1}$ are suggested by OMI data over India and Northern China over 2005–2016, most likely due to rising anthropogenic emissions. Over the last years, however, emissions appear to stabilize over India, and even decline over the North China Plain, as result of emission
regulations. Over Southeastern U.S., the observed decline of summertime OMI columns might have multiple causes, including the decline of $\mathrm{NO_x}$ levels (and therefore of the HCHO yield from VOC oxidation) and a decline of isoprene emissions due to $CO_2$ increase and land use change, both of which potentially accounting for $\sim -0.4\%\,\mathrm{yr}^{-1}$ to the isoprene emission trend according to current estimates.

*Data availability.* The QA4ECV dataset can be found at http://doi.org/10.18758/71021031 (De Smedt et al., 2017). The NASA aircraft
campaign datasets are available from the Langley Research Center at https://www-air.larc.nasa.gov/missions/merges. The HCHO FTIR data can be requested from the PIs of each station. The MEGAN-MOHYCAN isoprene inventory is available at https://emissions.aeronomie.be.

*Author contributions.* JFM coordinated the study and wrote the manuscript. JFM and TS designed and prepared the inversions. JFM and TS carried out the analysis, with help from GMO, BO and AG on specific aspects. IDS described the TROPOMI HCHO data. CV provided the



description of the FTIR dataset. CV and BL conducted the OMI validation and bias-characterization using FTIR data. BL, CV, CA, MG, JH,
FH, RK, EL, EM, MM, JMM, IMo, IMu, TN, JN, IO, MP, AR, WS, KS, RS, and YT provided the FTIR measurements. AF provided the
CAMS, TDLAS and DFGAS in situ measurements. All authors read and commented on the manuscript.

*Competing interests.*   The contact author has declared that none of the authors has any competing interests.

*Acknowledgements.*   HCHO satellite data from OMI were produced in the scope of the European FP7 project QA4ECV (grant no. 6007405).
We thank Dan Smale and the National Institute of Water and Atmospheric Research (NIWA) for the provision of the Lauder FTIR data, and
Nicholas Jones of the Centre for Atmospheric Chemistry, University of Wollongong, Wollongong, Australia, for provision of Wollongong
FTIR data. Thanks to Thomas Hanisco and the NASA Goddard Space Flight Center (GSFC) for provision of ISAF data (SENEX). B. O. was
supported by the EQUATOR (Emission Quantification of Atmospheric tracers in the Tropics using ObseRvations from satellites, contract
no. B2/202/P1/EQUATOR, 2021-2025) project of the BRAIN-be 2.0 programme from the Belgian Science Policy Office (Belspo). G.-M. O.
was supported by the SEEDS project funded by the European Commission under the H2020 programme (grant agreement no. 101004318).
C. V. was supported by Belspo through the ProDEx project TROVA-E2 (TROPOMI Validation - Phase E2) of the European Space Agency
(2020-2023). E. M. is a senior research associate with the F.R.S. - FNRS.

The Paris site has received funding from Sorbonne Université, the French research center CNRS, the French space agency CNES, and
Région Île-de-France. FTIR operations of Rikubetsu and Tsukuba are supported in part by the GOSAT series project. The Rikubetsu NDACC
site is funded by the joint research program of the Institute for Space-Earth Environmental Research (ISEE), Nagoya University. This work
has been supported by the BMBF (German Ministry of Research and Education) in the project ROMIC-II, subproject TroStra (01LG1904A).
We thank the AWI Bremerhaven, Germany and the Meteorological Service Suriname for logistical and the senate of Bremen for logistical
and financial support. We acknowledge the support of the station personnel at the AWIPEV research base in Ny-Ålesund, Spitsbergen and
C. Becker for support in Paramaribo, Suriname.



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
