# Peer review of "Bias correction of OMI HCHO columns based on FTIR and aircraft measurements and impact on top-down emission estimates"

_EGUsphere, 2023_

## Referee Comment (RC2)

**Review of "Bias characterization of OMI HCHO columns based on FTIR and aircraft measurements and impact on top-down emission estimates" by Muller et al., ACP, 2023**

**Description:**
The authors bias correct global OMI HCHO observations for 2005 to 2016 using ground-based total column density observations of HCHO from a global network of FTIR instruments and continuous in situ aircraft observations of HCHO from aircraft campaigns over the US. The bias corrected columns are used to derive biomass burning, biogenic and anthropogenic NMVOCs emissions and the results are compared to widely used bottom-up emissions inventories and to past estimates obtained with HCHO column densities and with satellite observations of isoprene concentrations. The authors also quantify trends in NMVOCs emissions and HCHO column densities. The manuscript is mostly clearly written (exceptions detailed below) and is suitable for publication in ACP after responding to concerns listed below.

**General Comments:**
The manuscript title is misleading, as "bias characterization" suggests that individual causes for biases in the OMI HCHO product will be diagnosed and quantified, but it is really a "bias correction" informed by independent observations. The manuscript and Sections 4.1 and 4.2 should be renamed to reflect this.

Section 2 should include details of the CrIS isoprene and Bauwens et al. (2016) datasets that the authors compare to in Section 5.3.

Is there any dependence of the sampling coincidence (50 km) on the regression statistics, as was found by Pinardi et al. (2020) for $NO_2$?

The updated / optimized model is evaluated against the same observations that are used to bias correct the OMI HCHO product used to derive emissions. The limitations of this evaluation should be acknowledged, given that it isn't truly independent.

The description of the OMI HCHO product in Section 2.1 is challenging to follow without prior detailed knowledge of the product. Non-European readers won't necessarily know what "EU-FP7" is. It's not apparent what the "cloud correction" is and what implication this has on the data. Defining what this is might help clarify what is meant by "in lieu of the cloud-corrected AMFs". Is the cloud fraction geometric or effective or something else? It's not clear what's being compensated for with model data over the Equatorial Pacific. Is the background removed in the background correction and the model data used to add back a background? Is the Equatorial Pacific the "reference region"?

It's not stated what fit is used to quantify trends in Section 5.4 and which of the reported trend values in Figures 12 and 14 are statistically significant.

The discussion of emissions trends in Section 5.4 has quite limited discussion of trends in biomass burning and the influence this has on trends in HCHO. For Africa, for example, Andela and van der Werf (2014) reported significant trends in biomass burning activity in

Africa and Hickman et al. (2021) reported decline in $NO_2$ abundances in North Equatorial Africa.

The Conclusion reads like a mix of concluding statements and as a discussion, as indicated by inclusion of citations.

**Specific Comments:**

L. 13: Consider a more informative summary of the comparison to CrIS isoprene than "striking similarities and differences".

L. 50: Should "higher-quality" be "higher spatial resolution"? If not, then perhaps indicate what it is about OMI that makes it higher quality.

L. 180-181: Justification for not using INTEX-B seems to contradict mention in Section 2.1 of the importance of the Pacific Ocean where OMI HCHO data are used to perform a background correction.

Figure 2: Intercept value and error estimate should be written to account for the scale of the axis ($10^{16}$).

Figure 12 caption: Unnecessary to include units in caption, as these are given in the figure.

L. 537-538: Clarify why a change in the version of the model would cause a change in data for 2017 and 2018? Is the updated TM5 model only applied to those years, rather than the full record being reprocessed to use the updated TM5 model?

L. 543-544: The statement starting "comparatively slower..." is confusing, as isoprene emissions have an exponential dependence on temperature, so should have a large response to warming. Is this statement meant to convey something else?

L. 550: Isn't 2012 too early for emissions to level off in India as a result of policies? Emissions controls were only implemented in earnest relatively recently, starting with power plants in 2015 and vehicles in 2018 (see for example Vohra et al., 2021)?

L. 564: Replace "is believed to be" with "one of"

L. 599: "poor" correlation would be R < 0.4.

L. 620-621: What's the utility of the sentence starting "This result demonstrates ...". It's an obvious statement. Is this stated because the product used by Bauwens et al. (2016) didn't undergo any validation?

**References:**

Andela and van der Werf, 2014, http://www.nature.com/doifinder/10.1038/nclimate2313
Hickman et al., 2021, https://doi.org/10.1073/pnas.2002579118
Pinardi et al., 2020, https://doi.org/10.5194/amt-13-6141-2020
Vohra et al., 2021, https://doi.org/10.5194/acp-21-6275-2021

---

## Author Comment (AC1)

**Reply to Reviewer#1**

We thank the referee for the constructive comments on our manuscript. We have made changes to the manuscript based on the recommendations of both referees. The responses to the referee's comments are provided below.

**The paper is extremely well written and thorough and addresses a topic of scientific importance. I appreciate the authors' careful analysis and forthright discussion of uncertainties. I strongly recommend publication and have only minor comments and suggestions.**

We thank very much the reviewer for the appreciation of our manuscript.

**1) The authors discuss inconsistency and uncertainty in the aircraft HCHO measurements used for satellite evaluation, and come up with a defensible approach for dealing with this. It would be helpful somewhere later on to include a brief discussion of the degree to which these uncertainties could impact (or not) their conclusions about OMI and the resulting emissions.**

Thanks for the very good point. The aircraft-based validation relies on two instrumental techniques: DFGAS and CAMS. The intercomparison of DFGAS and CAMS suggests a potential overestimation of DFGAS by up to 11% and a potential bias of the adjusted CAMS measurements (see Sect. 2.3) of up to ca. ±6.5%. Assuming that the optimized model columns are proportional to the observed mixing ratios, we evaluated the impact of those potential biases on the regression statistics. The regression slope ranges between 0.6 and 0.7, while the intercept lies in the range $(2.6\text{-}3.3)\times10^{15}$ molec. cm$^{-2}$. We added this discussion to Sect. 4.2.

**2) There is a sensitivity inversion (OPT3) included to assess impacts of the prior error assumptions on the inversion results, which I agree is an important test to include. However, the results and conclusions from this test do not seem to be discussed anywhere.**

The results of the comparison with aircraft data with OPT3 were summarized in Table S4 in the supplement, but we agree with the referee that the OPT3 results should have been discussed in the main manuscript. The point is, that the OPT3 run are not very different from those of the OPT2 run. For example, the optimized isoprene emissions of OPT3 are only slightly higher (by ~3%) than in OPT2. The statistics of the comparisons with aircraft data (in 2012 and 2013) are also quite similar. The Sect. 5.1 and 5.3 were updated to convey this information.

**3) 281-290, I am confused here b/c the text first says that E is diagonal but then later the text describes a decorrelation length scale which seems to imply the presence of off-diagonal elements. Please clarify.**

Thank you for the useful comment. There was a typo on line 287 where the matrix **E** should have been the matrix **B**. The matrix **E** is diagonal, whereas the matrix **B** is not.

**4) 543-544, "and comparatively slower increases (< 1%yr$^{-1}$) in biogenic VOC emissions over many areas due to global warming (e.g. over Amazonia, Southern Africa and Australia) (Fig. 14)."**

**The phrasing here implies that warming is unequivocally driving a statistically significant, detectable increase in emissions over these regions. Looking at the figure, however, the trend for Amazonia is 0.0%/y and I have a hard time believing that the trends for Southern Africa and Australia are statistically distinguishable from zero. I wonder if what the authors mean to say is that any warming driven isoprene increase is small or undetectable over this period; that is the conclusion I draw from Fig. 14.**

We did not mean that warming induces detectable emission increases over all those regions; instead, we meant that warming might cause emission increases that do not exceed 1% yr$^{-1}$. The uncertainties on the trends are now indicated on Fig. 12 and 14. The increasing emission trend (Fig. 14) is not significant over Amazonia and Southern Africa and it is barely significant over Australia. We therefore changed as follows the sentence pointed out by the referee: "and comparatively slower changes (< 1% yr$^{-1}$) in biogenic VOC emissions over many areas (e.g. over Amazonia, Southern Africa and Australia) (Fig. 14)."

**5) Sections 3.3-3.4, I get the impression that emissions are being optimized on a monthly basis but I don't believe this is explicitly stated. Please specify.**

The emissions are indeed optimized on a monthly basis. This is now made clear in Sect. 3.3.

**6) In lines 515-520 the authors discuss the difficulty in separating biogenic versus pyrogenic emissions in some regions. I feel that the paper would benefit from a more general discussion of this issue, perhaps earlier on when introducing the inversion methodology. That is, we are solving for 3 separate variables (anthropogenic, biogenic, and pyrogenic VOC emissions) for every grid cell based on a single observed variable (HCHO). To what degree are these terms actually resolved through the inversion, and to what degree does that separation merely rely on the prior and/or only work where (again according to the prior) one source is dominant?**

Fair point. We have added the following paragraph in Section 3.4:

*The determination of VOC emissions from satellite HCHO data has several limitations. Although the fluxes from three emission categories are inverted simultaneously through the minimization of the cost function (Eq. 1), the distinction between these categories is uncertain, in particular at places and times where more than one category is dominant. The optimization realizes the separation largely based on the a priori magnitude and spatio-temporal patterns of the emissions, through the correlation between a priori errors on the emission parameters. Therefore, errors in the a priori emission distributions might cause errors in the attribution of emissions between different categories. Fortunately, a single emission category is very often dominant over continental areas, e.g. anthropogenic emissions are strongly dominant over northeastern China and biogenic emissions are dominant over Eastern U.S. and most tropical forests. However, biomass burning is a highly episodic source which generally coincides with biogenic source areas, resulting in uncertain top-down emissions for both biogenic and pyrogenic sources. The same is also true in areas (e.g. India) where both anthropogenic and biogenic emissions are significant. In those regions, the total top-down VOC emissions are much better resolved than individual categories.*

**7) The VOC source optimization is by nature indirect and based on the resulting HCHO abundance. The authors should include some assessment or discussion of the extent to which the VOC emission magnitude updates could in fact be compensating for other factors or model errors that affect HCHO (for example, incorrect VOC speciation, uncertainty in the diel cycle of VOC emissions, errors in NOx emissions or in the HCHO lifetime, uncertainties in the chemistry leading to HCHO, etc.).**

We added the following paragraph in Section 3.4:

*In addition, the top-down VOC emissions have uncertainties related to the multiple factors that might affect the abundance of HCHO, besides the magnitude of the emissions. This includes, for example, the background HCHO abundance, largely determined by OH radical levels, the incomplete and oversimplified speciation of VOCs in large-scale models, the VOC chemical oxidation mechanisms, the deposition of VOC oxidation intermediates, the diurnal cycle of emissions, especially for biomass*

*burning, the vertical transport processes control the vertical profile of HCHO, and the $NO_x$ levels, which influence the yields of HCHO from many important VOCs. Although a few of those uncertainties were partially addressed in previous studies (e.g. Oomen et al., 2023), an exhaustive quantitative study of those uncertainties would be a daunting task and is beyond the scope of the present study.*

---

## Author Comment (AC2)

**Reply to Reviewer#2**

We thank the referee for the constructive comments on our manuscript. We have made changes to the manuscript based on the recommendations of both referees. The responses to the referee's comments are provided below.

**The manuscript title is misleading, as "bias characterization" suggests that individual causes for biases in the OMI HCHO product will be diagnosed and quantified, but it is really a "bias correction" informed by independent observations. The manuscript and Sections 4.1 and 4.2 should be renamed to reflect this.**

Thank you for the suggestion. The manuscript has been amended accordingly.

**Section 2 should include details of the CrIS isoprene and Bauwens et al. (2016) datasets that the authors compare to in Section 5.3.**

We added a new subsection on CrIS isoprene in Section 2. We do not think necessary to do the same for the Bauwens et al. dataset since the nature and main results of that study are already sufficiently described in the manuscript.

**Is there any dependence of the sampling coincidence (50 km) on the regression statistics, as was found by Pinardi et al. (2020) for NO₂?**

Thanks for this comment. Note that, compared to $NO_2$, HCHO columns show generally much less spatial heterogeneity around cities due to anthropogenic emissions. We performed tests with different collocation criteria and added the following text in Sect. 4.1:

*We tested alternative choices for the collocation distance: a higher value (100 km instead of 50 km) degrades the correlation and yields a slightly lower slope (0.63) than our reference regression. Lower distances (e.g. 20 km) lead to an excessively low number of OMI pixels to be averaged and therefore to poor correlation with FTIR.*

**The updated / optimized model is evaluated against the same observations that are used to bias correct the OMI HCHO product used to derive emissions. The limitations of this evaluation should be acknowledged, given that it isn't truly independent.**

This comment is not correct. The updated model is evaluated against many more aircraft campaign data than the observations used to bias correct the OMI product. See Figure 1 and Table 1.

**The description of the OMI HCHO product in Section 2.1 is challenging to follow without prior detailed knowledge of the product. Non-European readers won't necessarily know what "EU-FP7" is. It's not apparent what the "cloud correction" is and what implication this has on the data. Defining what this is might help clarify what is meant by "in lieu of the cloud-corrected AMFs". Is the cloud fraction geometric or effective or something else? It's not clear what's being compensated for with model data over the Equatorial Pacific. Is the background removed in the background correction and the model data used to add back a background? Is the Equatorial Pacific the "reference region"?**

The QA4ECV EU-FP7 project is now described as "QA4ECV project of the 7th Framework Programme of the European Union (EU-FP7)". The description of the algorithm has been updated as follows:

(…) *The standard AMF calculations uses the effective cloud fraction and cloud top pressure from the Fresco v7 cloud product (Veefkind et al., 2016), treating clouds as Lambertian reflectors and applying the independent pixel approximation (Martin et al., 2002; Boersma et al., 2004). However, in this work,*

*the cloud correction is switched off, except for a strict filtering (effective cloud fraction > 0.2). Clear-sky air mass factors (AMF) are used in lieu of the cloud-corrected AMFs. This choice ensures an optimal consistency with the TROPOMI HCHO dataset (De Smedt et al., 2021). Indeed, the TROPOMI HCHO retrieval is inherited from the QA4ECV algorithm with the aim to generate a consistent time series of early afternoon observations. Finally, to correct for any global offset and for stripes arising between the rows, a background correction is performed on daily basis using the HCHO slant columns over the Pacific Ocean. The TM5 HCHO model columns derived in the same region are finally added to compensate for the background HCHO concentrations. (…)*

**It's not stated what fit is used to quantify trends in Section 5.4 and which of the reported trend values in Figures 12 and 14 are statistically significant.**

This is a good point. We now make clear that we use a least squares linear regression method. The 1-σ errors of the regression slopes are now given in Figures 12 and 14.

**The discussion of emissions trends in Section 5.4 has quite limited discussion of trends in biomass burning and the influence this has on trends in HCHO. For Africa, for example, Andela and van der Werf (2014) reported significant trends in biomass burning activity in Africa and Hickman et al. (2021) reported decline in $NO_2$ abundances in North Equatorial Africa.**

Thank you for this interesting comment. Note that the a priori inventory of biomass burning emissions (GFED4s) takes into account the reported trends in biomass burning activity, e.g. the emission decline in the savanna region of northern Africa (Hickman et al., 2021). The patterns of changes over Africa are complex (see Fig. 13) and a dedicated investigation would be needed to elucidate their causes. Nevertheless, the manuscript now briefly discusses the role of biomass burning as an important driver for long-term trends in HCHO, in particular over Africa.

**The Conclusion reads like a mix of concluding statements and as a discussion, as indicated by inclusion of citations.**

The referee is correct. We simplified the Conclusion section and removed most citations. The bits of discussion that are now removed from the Conclusions are part of the general text.

**Specific Comments:**

**L. 13: Consider a more informative summary of the comparison to CrIS isoprene than "striking similarities and differences".**

For the sake of brevity, we prefer to keep the sentence as is. The details are provided in Sect. 5.3.

**L. 50: Should "higher-quality" be "higher spatial resolution"? If not, then perhaps indicate what it is about OMI that makes it higher quality.**

Fair point. We changed to "higher-resolution sounders".

**L. 180-181: Justification for not using INTEX-B seems to contradict mention in Section 2.1 of the importance of the Pacific Ocean where OMI HCHO data are used to perform background correction.**

Our global inversion study addresses continental VOC emissions, and, as explained in the text, HCHO levels over the Pacific are insensitive to emissions over land. Therefore there would be no point in validating the inversion results using such data.

**Figure 2: Intercept value and error estimate should be written to account for the scale of the axis ($10^{16}$).**

Although strictly speaking, this comment is of course correct, we keep the figure as is, since it is obvious for the reader that the intercept should not be multiplied by the scale factor of $10^{16}$.

**Figure 12 caption: Unnecessary to include units in caption, as these are given in the figure.**

Corrected as proposed.

**L. 537-538: Clarify why a change in the version of the model would cause a change in data for 2017 and 2018? Is the updated TM5 model only applied to those years, rather than the full record being reprocessed to use the updated TM5 model?**

The TM5 model changes were not applied to the full record. There are two different versions of TM5 used for the QA4ECV OMI HCHO product. The TM5 version which was run during the QA4ECV project, and the TROPOMI version (TM5-MP) from 2018 onwards. There is no temporal overlap between the two versions. TM5-MP has not been run for the years before 2018. Reasons for change in HCHO profiles are not clear (possibly the time resolution, the definition of the vertical layers, or an update in the convection scheme). They are apparent only in Tropical regions.

**L. 543-544: The statement starting "comparatively slower..." is confusing, as isoprene emissions have an exponential dependence on temperature, so should have a large response to warming. Is this statement meant to convey something else?**

No, the statement is meant to convey that over the large regions shown on Fig. 12, global warming does not induce fast HCHO trends (>1%/yr). Even with an exponential temperature-dependence of biogenic emissions, a fast warming trend is require to induce a large trend in HCHO columns. Although such warming trends have been observed at many locations (see Stavrakou et al., 2018), spatial averaging over large regions leads to moderate trends, as observed on Fig. 12.

**L. 550: Isn't 2012 too early for emissions to level off in India as a result of policies? Emissions controls were only implemented in earnest relatively recently, starting with power plants in 2015 and vehicles in 2018 (see for example Vohra et al., 2021)?**

Thank you for this interesting comment. We changed the manuscript and included the following text:

*Over India, the apparent stabilization of top-down emissions after 2012 seems contradicted by reports that regulatory measures were not effective in India until the last years (after 2018) (Vohra et al., 2021). More work will be needed to examine the patterns of HCHO changes and the possible causes of the discrepancy.*

**L. 564: Replace "is believed to be" with "one of"**

We changed to "is the dominant source". Isoprene oxidation is not just one of the sources, it is the dominant source in this region.

**L. 599: "poor" correlation would be R < 0.4.**

It's all relative. But we changed to "low" instead of "poor".

**L. 620-621: What's the utility of the sentence starting "This result demonstrates ...". It's an obvious statement. Is this stated because the product used by Bauwens et al. (2016) didn't undergo any validation?**

Yes indeed, as implied in the manuscript (e.g. the Introduction), previous inverse modelling studies did not apply bias correction to the HCHO datasets. It might be obvious that bias-corrected is better than not, but it is not obvious that it has such large consequences on top-down emissions.

**References:**

**Andela and van der Werf, 2014, http://www.nature.com/doifinder/10.1038/nclimate2313**

**Hickman et al., 2021, https://doi.org/10.1073/pnas.2002579118**

**Pinardi et al., 2020, https://doi.org/10.5194/amt-13-6141-2020**

**Vohra et al., 2021, https://doi.org/10.5194/acp-21-6275-2021**

Boersma et al., 2004, https://doi.org/10.1029/2003JD003962

Martin et al., 2002, https://doi.org/10.1029/2001JD001027

Stavrakou et al., 2018, https://doi.org/10.1029/2018GL078676

---

## Author Response (AR2)

Dear Editor,

Thank you for the positive evaluation of the manuscript updates. Please find hereafter the responses to your and Reviewer#2's comments. We hope that those responses will be considered satisfactory.

Best regards,

Jean-François Müller
* * *
**Reviewer#2**

**(1) For my original comment starting "The updated / optimized model is evaluated against the same observations that are used to bias correct the OMI HCHO product used to derive emissions.", the authors point out that this is not correct, so I'd recommend making this clearer to the reader and in so doing this more effectively convey that the evaluation is independent. Figure 1 and Table 1 are clear, but these do not convey that more observations are used in the evaluation than in the bias correction.**

The Figure 1 legend clearly distinguishes the aircraft campaigns used for the bias correction and the additional campaigns used for model evaluation. Nevertheless we changed the main text with the following sentence "Those campaigns, as well as 7 additional aircraft campaigns (datasets 5-11 in Table1) are used to evaluate the inverse modelling results. "

**(2) For my comment starting "The description of the OMI HCHO product in Section 2.1 is ... ", the updated text provided by the authors still doesn't convey what this cloud correction is. It would help to state that this cloud correction is ordinarily applied to the AMF.**

Thanks for this comment, we updated the text as follows

"However, in this work, the cloud correction to the AMF calculation is switched off..."

**(3) Related to the same original comment in (2) above, the updated text tells us that the background values from the model are finally added, but missing from this is "finally added to the vertical columns of HCHO", if this is the case.**

Updated as suggested.

**Editor**

**In addition, in lines 125-126, you mention about the row anomaly. Can you please comment on this in Conclusion, whether we can you use the OMI data after the year 2014?**

As explained in Section 2.1, there is a gradual degradation of the spatial coverage. However it does imply a degradation of the quality of the data, even after 2014. However we did note in Section 5.4 a deterioration of the satellite data after 2016, possibly related to instrumental degradation and/or changes in the TM5 model used to compute the air mass factors. A sentence was added in the conclusions.